# In vivo RNA interactome profiling reveals 3'UTR-processed small RNA targeting a central regulatory hub

Fang Liu[1,2,7], Ziying Chen[1,3,4,7], Shuo Zhang[1,2,5], Kejing Wu[1,2], Cheng Bei[3], Chuan Wang [3] ✉ & Yanjie Chao [1,2,5,6] ✉

Small noncoding RNAs (sRNAs) are crucial regulators of gene expression in bacteria. Acting in concert with major RNA chaperones such as Hfq or ProQ, sRNAs base-pair with multiple target mRNAs and form large RNA-RNA interaction networks. To systematically investigate the RNA-RNA interactome in living cells, we have developed a streamlined in vivo approach iRIL-seq (intracellular RIL-seq). This generic approach is highly robust, illustrating the dynamic sRNA interactomes in *Salmonella enterica* across multiple stages of growth. We have identified the OmpD porin mRNA as a central regulatory hub that is targeted by a dozen sRNAs, including FadZ cleaved from the conserved 3'UTR of *fadBA* mRNA. Both *ompD* and FadZ are activated by CRP, constituting a type I incoherent feed-forward loop in the fatty acid metabolism pathway. Altogether, we have established an approach to profile RNA-RNA interactomes in live cells, highlighting the complexity of RNA regulatory hubs and RNA networks.

Gene regulation at the post-transcriptional level is crucial for the rapid, global response to environmental changes and survival under stressful conditions. The largest class of post-transcriptional regulators in bacteria are small noncoding RNAs of 50–250 nt in length[1–3]. Bacterial sRNAs often act in concert with major RNA chaperones such as Hfq or ProQ[4], and regulate the expression of target mRNAs via direct base-pairing interactions[2]. Most sRNAs regulate more than one mRNA using a conserved seed sequence that has partial complementarity to target mRNAs[5–7]. Vice versa, functionally important mRNAs (such as the general stress regulator *rpoS* and biofilm regulator *csgD*) are targeted by several different sRNAs, which have been conceptualized as mRNA regulatory hubs[8]. These mRNAs and regulatory sRNAs thus constitute a large and intricate RNA-RNA interaction network involving hundreds of genes in bacteria[9,10]. The scale of this RNA-RNA network now begins to rival that of the protein-protein and protein-DNA interaction networks, and covers all aspects of bacterial physiology including central metabolism, cell shape, envelope integrity, quorum sensing, biofilm formation, antibiotic resistance, host infection, symbiosis and more[11–18].

The bacterial RNA interaction network has been rapidly expanding, driven by the discovery of novel sRNAs from unexpected genomic locations[3,19–21], and also by profiling the RNA interactome using global approaches[10,22–24]. RIP-seq (RNA immunoprecipitation and sequencing) profiling of Hfq-bound transcripts have discovered a plethora of sRNAs that are derived from the 3' region of mRNAs[20,25]. These sRNA are either transcribed from gene-internal promoters[20,26], or processed from

[1]Microbial RNA Systems Biology Unit, Key Laboratory of Molecular Virology and Immunology, Shanghai Institute of Immunity and Infection, Chinese Academy of Sciences, Shanghai 200031, China. [2]The Center for Microbes, Development and Health (CMDH), Institut Pasteur of Shanghai, Chinese Academy of Sciences, Shanghai 200031, China. [3]Key Laboratory of Medical Molecular Virology (MOE/NHC/CAMS), Shanghai Frontiers Science Center of Pathogenic Microorganisms and Infection, School of Basic Medical Sciences, Fudan University, Shanghai 200033, China. [4]Department of Colorectal Surgery, Fudan University Shanghai Cancer Center & Department of Oncology, Shanghai Medical College, Fudan University, Shanghai 200032, China. [5]University of Chinese Academy of Sciences, Beijing 100049, China. [6]Key Laboratory of RNA Science and Engineering, Shanghai Institute of Biochemistry and Cell Biology, Center for Excellence in Molecular Cell Science, Chinese Academy of Sciences, Shanghai 200031, China. [7]These authors contributed equally: Fang Liu, Ziying Chen. ✉e-mail: chuanwang@fudan.edu.cn; yjchao@ips.ac.cn

mRNAs by the major endoribonuclease RNase E[12,15,27–30]. The latter was first exemplified by the CpxQ sRNA, a 58-nt 3′UTR fragment cleaved from the mRNA of envelope-stress chaperone CpxP and represses the expression of multiple extracytoplasmic proteins involved with envelope stress response[27,31]. Despite the absence of a 5′PPP cap, these processed 3′UTR-sRNAs stably accumulate in vivo and function as missing regulatory arms in well-characterized pathways[21]. With hundreds of *Salmonella* mRNAs that have the potential to produce 3′UTR-derived sRNAs by RNase E cleavage[25], the true number of such 3′UTR-sRNAs may be underestimated, leaving key regulatory nodes missing in the RNA interaction network and physiological pathways.

To interrogate the bacterial RNA-RNA interactome, several global approaches have been developed to profile sRNA-mediated interactions, such as RIL-seq (RNA-interaction by ligation and sequencing[23]), CLASH (crosslinking, ligation, and sequencing of hybrids[24]), and Hi-GRIL-seq (high-throughput global sRNA target identification by ligation and sequencing[22,32]). Both RIL-seq and CLASH rely on UV-crosslinking and in vitro RNA ligation using a purified T4 RNA ligase, which mediates proximity-ligation of sRNAs to their interacting partners crosslinked on an RNA-binding protein such as Hfq or ProQ. It has enabled systems-wide analyses of RNA-RNA interactomes in several Gram-negative bacterial pathogens including *E. coli*[23,33,34], *S. enterica*[35], *Pseudomonas aeruginosa*[36], and *Vibrio cholerae*[13]. These global analyses identified new sRNAs acting as RNA sponges, advanced our understanding of sRNA-target interactions, and revealed fundamental regulatory networks. However, current approaches are complicated by the use of irreversible UV crosslinking and lengthy in vitro on-bead reactions including RNase trimming, RNA end repair, ligation, and protease digestion steps prior to cDNA library preparation[37]. These technically demanding and error-prone steps may introduce in vitro artefacts and biases, hindering the identification of physiologically important regulations in vivo and the elucidation of RNA interactomes in living cells. Hi-GRIL-seq is an alternative in vivo approach that enables RNA proximity-ligation by T4 RNA ligase expressed inside bacteria[32], which however has noisy background and limited throughput for in-depth analysis of RNA interactome at the systems-wide level.

Herein, we have developed a novel in vivo approach to profile RNA-RNA interactomes in living cells (Fig. 1a), harnessing in vivo proximity-ligation followed by RIP-seq to enrich the ligation products. Because this approach does not require any crosslinking or enzymatic digestions in vitro, it is highly streamlined and can be finished within a single day before RNA sequencing. We have demonstrated its high performance and robustness by interrogating the dynamic sRNA interactome during *Salmonella* growth across several different stages. Strikingly, we have identified the porin-encoding mRNA *ompD* as a central regulatory hub that is targeted by an unprecedented number of twelve sRNAs, including a novel 3′UTR-derived sRNA named FadZ. Our data show that FadZ is processed from the conserved 3′UTR of *fadBA* mRNA encoding fatty acid oxidation enzymes, and represses the expression of OmpD via direct base-pairing interactions. Both FadZ and *ompD* are activated by the same transcription factor CRP, constituting a critical feed-forward loop in the fatty acid metabolism pathway.

## Results

### iRIL-seq profiles RNA-RNA interactome in living cells

To systematically profile the RNA-RNA interactome in live bacteria, we have developed a highly streamlined in vivo approach (iRIL-seq, intracellular RNA interaction by ligation and sequencing) (Fig. 1a). By pulse-expressing T4 RNA ligase 1 (*t4rnl1*) from an inducible pBAD promoter (Fig. 1a, Supplementary Fig. 1), iRIL-seq enables in vivo proximity ligation of sRNAs to their interaction partners in living cells. This is followed by enrichment of Hfq-bound ligation products (RNA chimeras) using Hfq-coIP and subsequent RNA-seq analysis[20].

Expression of T4 RNA ligase was induced only for 30 min, minimizing non-specific ligations and secondary effects on *Salmonella* growth (Supplementary Fig. 1c).

To validate the feasibility of the approach, we have successfully detected in vivo ligation products using RT-PCR for several known sRNA-target pairs in Hfq-coIP samples (Supplementary Fig. 1d–f), such as ArcZ-*flhD*[38] and CyaR-*ompX*[39]. As important controls, chimeras were not detected in the absence of T4 RNA ligase or in the control IP (untagged WT) samples. We also did not detect chimeras for non-target interactions (e.g., ArcZ-*ompX*, which are not predicted to base-pair), together indicating the high specificity and sensitivity of our in vivo ligation and capture strategy.

RNA-seq analysis of the Hfq-coIP samples on a genome-wide scale fully recapitulated the RNA ligation products for ArcZ-*flhD* in the form of chimeric reads (Fig. 1b, c). Systematic analysis of all sequencing reads (chimeras and non-chimeric singletons) confirmed a strong enrichment of known Hfq-associated sRNAs in Hfq-coIP samples vs. the untagged control coIP library (Fig. 1c), but also a 10-fold increase in the number of chimeric reads when T4 RNA ligase was expressed (e.g., Hfq + T4 vs. Hfq + EV) (Fig. 1d). Moreover, while the majority of the detected chimeras in the no-ligase control (Hfq + EV) represented ligation products of abundant rRNAs and tRNAs, a large number of 'informative' non-rRNA/tRNA chimeras (8000 chimeras/million reads) were detected in the Hfq + T4 samples (Fig. 1c, d). Further analysis of the significant chimeras (S-chimeras, Fisher's-exact test, $p < 0.05$) confirms that most sRNAs are ligated to the 5′UTR and CDS of mRNAs (Fig. 1e), consistent with the established action of sRNAs[2]. Within these S-chimeras, nearly all mRNA 5′UTRs and CDS are located at the 5′ end ("RNA1"), and over 90% of sRNAs are located at the 3′ end ("RNA2") (Fig. 1f). This directionality indicates that sRNAs 5′ are prone to in vivo ligation, whereas sRNA 3′ ends are protected by Hfq from fortuitous ligation. These data not only fully recapitulate the previous RIL-seq results[23,40], and also support the high fidelity of in vivo proximity-ligation on Hfq in live bacteria.

### Global RNA-RNA interaction network overgrowth in *Salmonella*

Having established the iRIL-seq approach, we next analyzed the global RNA interactome at three different stages of *Salmonella* growth in LB medium. At EP (exponential phase), ESP (early stationary phase) and SP (stationary phase), we induced the expression of T4 RNA ligase in *Salmonella* for 30 min, and then pulled down Hfq and its bound RNAs (Fig. 2a). Deep sequencing analysis of the RNA samples (12 samples, in duplicates) yielded ~153 million high-quality mappable reads (Supplementary Data 4), generating ample coverage of the *Salmonella* genome with high reproducibility between replicates (Supplementary Fig. 2a, b). iRIL-seq strongly enriched the class of Hfq-associated sRNAs (Fig. 2b, Supplementary Fig. 2c), which exhibited a dynamic profile overgrowth consistent with our previous report[20]. Further, we have observed a strong enrichment of sRNA chimeras at all three growth conditions (Fig. 2b), covering ~90% known Hfq-dependent sRNAs. Using the RIL-seq computational pipeline with a stringent cutoff (chimeric reads ≥ 10, and $p < 0.05$, Fisher's-exact test[23]), our iRIL-seq analysis identified a total of 436, 855 and 1705 statistically significant RNA–RNA interactions under three growth conditions, respectively (Fig. 2c). Nearly 30% of these interactions were consistently detected in more than one growth condition. Interestingly, the number of interactions increased towards stationary phase of growth (Fig. 2c–e), accompanied by the appearance of stress-induced sRNAs and their abundant interactions with target genes (Fig. 2b). For example, 10-fold more interactions were identified for SdsR in SP compared to EP (Fig. 2d, f), when SdsR was activated by RpoS and occupied ~20% of all sRNA singleton reads (Fig. 2b). Similarly, the stress-related sRNA RprA, an activator of RpoS[41], showed 20-fold more interactions in stationary phase, which included the known RprA-*rpoS* interaction (Supplementary Data 5). Therefore, iRIL-seq analysis established a comprehensive

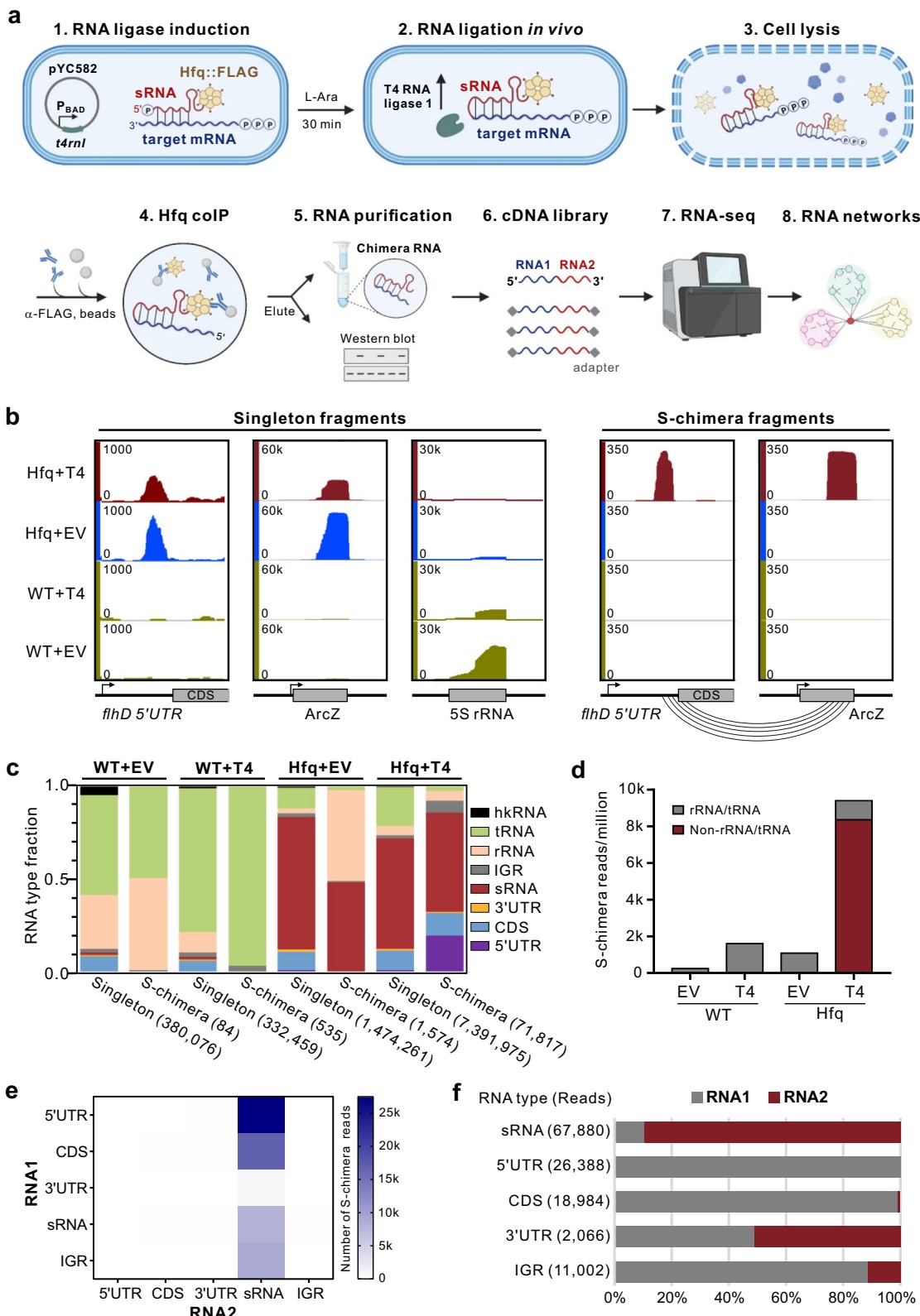

and dynamic sRNA-target interaction network in vivo during *Salmonella* growth.

## iRIL-seq detects targets for both primary and processed sRNAs with high accuracy

Cross-comparing our in vivo data with available RIL-seq dataset from *Salmonella* under the same ESP condition[35] confirmed that iRIL-seq

performed equally well with a greatly streamlined workflow (Supplementary Fig. 3a–d). Using the Top 10 sRNAs as a practical benchmark, iRIL-seq and RIL-seq identified a similar number of interactions with a significant overlap (Fig. 3a). The RNA pairs in S-chimeras identified by iRIL-seq displayed a significantly lower hybridization energy than random RNAs, and also significantly lower than the S-chimeras identified by RIL-seq (Supplementary Fig. 3d). Meta-analysis of the

**Fig. 1 | iRIL-seq faithfully captures sRNA-target interactions. a** Schematic of iRIL-seq. *Salmonella* Hfq::FLAG strain carrying the plasmid pBAD-*t4rnl1* (pYC582) was grown in LB. sRNA-target pairs were ligated to form chimeras in vivo by T4 RNA ligase induced for 30 min with L-arabinose. The ligation chimeras bound to 3xFLAG tagged Hfq were enriched using coIP from bacterial lysates. Chimeras were then purified, identified by deep sequencing, and used to determine the RNA interaction network by subsequent in silico analysis. Created with BioRender.com. **b** iRIL-seq captured known sRNA-target interactions. Genome browser screenshots showing the genomic locations of indicated RNAs covered by singleton or significant chimera reads (S-chimera, *p* < 0.05, one-sided Fisher's exact test). *Salmonella* WT and Hfq::FLAG strains carrying empty vector or pYC582 were grown in LB. Bacteria were treated with L-arabinose for 30 min and grown to $OD_{600}$ of 2.0. Cells were collected and performed iRIL-seq. EV: empty vector. T4: pBAD-*t4rnl1* (pYC582). ORFs and RNAs were indicated by gray boxes. **c** Distribution of each transcript type for singleton and S-chimeric fragments within one set of four iRIL-seq libraries. The total number of sequenced fragments was denoted in parentheses. hkRNA: four housekeeping RNA (RnpB, SsrS, Ffs and SsrA). IGR: intergenic region. EV and T4 are the same as in (**b**). **d** Number of S-chimeric fragments detected in each sample. EV and T4 are the same as in (**b**). **e** Number of S-chimeric fragments for different transcript types. RNA1, the 5' terminal RNA in the chimera. RNA2, the 3' terminal RNA in the chimera. **f** Distribution of RNA1 and RNA2 in S-chimeras for different transcript types.

S-chimera distribution confirmed that RNA1 was mostly enriched at mRNA 5'UTRs near the translational start codons (Supplementary Fig. 3e, f), the canonical binding sites for most sRNAs[2].

When considering all three growth conditions, iRIL-seq identified nearly half of interactions (203) from RIL-seq[35], and another 1214 new interactions (Supplementary Fig. 3g). To further examine the reliability of these interactions, we carefully analyzed the S-chimeric reads for sequence motifs. This revealed a polyU motif in RNA2 (Fig. 3b), likely a signature of Rho-independent terminators at sRNA 3' ends (Supplementary Figs. 4, 5). Meta-analyses of RNA1 fragments successfully identified a number of highly significant motifs with extremely low *p*-values (Fig. 3c–f, Supplementary Fig. 6). Strikingly, these motifs are found in >95% of all the captured target mRNAs, and show substantial complementarity to the conserved seed region of their cognate sRNAs. For example, we have identified two motifs complementary to the both seeds (R1 & R2) of the GcvB sRNA (Fig. 3f)[42,43]. These data demonstrate that iRIL-seq is highly effective to discover true sRNA-target interactions in vivo.

In addition to these sRNA-mRNA interactions, we also identified almost 100 sRNA-sRNA chimeras involving many potential RNA sponges (Supplementary Data 5). For instance, the documented sponge interaction between ArcZ and CyaR sRNAs[33,44] had the highest abundance among ArcZ S-chimeras in our dataset. We also captured the classical ChiX-*chbBC* sponge pairs[45], as well as the OppX-MicF sponge complex that was recently recognized to adjust envelope porosity to transport capacity[35].

In live cells, the performance of iRIL-seq is based on the in vivo availability of sRNA 5' end for proximity ligation (Fig. 1a), since we have not introduced any nuclease trimming or end-repair steps. Intriguingly, we observed a strong enrichment of processed sRNAs in S-chimeras compared to primary sRNAs (Fig. 3g), indicating processed sRNAs with a 5'-monophosphate (5'P) are more prone to ligation. Indeed, the 5' ends of processed sRNA such as CpxQ and ArcZ are readily captured as RNA2 in S-chimeras and occupy a large number of chimeric reads (Fig. 3h). Based on this unique feature, iRIL-seq may help identify novel 3'UTR-processed sRNAs that are prone to ligation. In comparison, primary sRNAs such as Spot42 and GcvB are involved in chimera formation at internal seed regions (Fig. 3h), perhaps during the coupled decay of sRNA-target pairs[46]. Therefore, our data confirm that T4 RNA ligase mediates ligation between the 5'P end of sRNAs to their binding partners on Hfq. This intrinsic ligation mechanism enables iRIL-seq to capture numerous interaction partners for both primary sRNAs and processed sRNAs with high accuracy.

## iRIL-seq identifies the porin mRNA ompD as key regulatory hub

Focusing on the target genes, our inspection of RNA1 fragments in chimeric reads identified a number of potentially key regulatory hubs in *Salmonella* that are targeted by multiple sRNAs. Figure 4a depicts *Salmonella* mRNAs that may interact with four or more sRNAs based on our data. These mRNAs include the prominent regulatory hub *rpoS*, whose expression is activated by three sRNAs (ArcZ, RprA, and DsrA[47]), all of which were captured as S-chimeras in our dataset (Supplementary Data 5).

Among the most-targeted mRNAs are *ompD* and *ompC*, both encoding abundant porins on the *Salmonella* outer membrane. The *ompD* mRNA is predicted to interact with as many as 13 sRNA candidates (Fig. 4a, b), among which only two are established regulators of *ompD*: the global OMP repressor RybB[48] and the pathogenicity island-encoded sRNA InvR[49]. Twelve of these sRNAs are predicted to base-pair with the 5'UTR or early CDS of the *omp*D mRNA (Fig. 4b), whereas ArcZ is predicted to interact in the coding region (Supplementary Fig. 7). To validate these interactions and their regulatory functions, we cloned RybB as control and another 11 sRNAs into pZE12 vectors and constitutively expressed them in WT *Salmonella*. Strikingly, 8 out of 11 sRNAs strongly inhibited the expression of OmpD (Fig. 4c), while several sRNAs also repressed OmpC and OmpA to different extent. These data not only suggest *ompD* as one the largest mRNA regulatory hubs in bacteria (regulated by >10 sRNAs), but also showcase the reliability and robustness of iRIL-seq analysis.

## A novel 3'UTR-derived sRNA regulator FadZ

iRIL-seq data suggest that several of the novel regulators of OmpD are processed sRNAs, among which we selected one sRNA STnc790 (renamed FadZ) for detailed characterization. FadZ was initially described as a primary sRNA candidate in *Salmonella* using differential RNA-seq[50,51] and proposed as a 3'UTR-derived sRNA[52]. It is located within the 196 nt-long 3'UTR of the *fadBA* mRNA, which encodes enzymes involved in fatty acid oxidation and metabolism (Fig. 5a). Our iRIL-seq data show that only a short 3' terminal fragment of the mRNA, which possesses the highest sequence conservation among Enterobacteriaceae species including *E. coli* and *Yersinia* (Fig. 5c, d), was pulled down by Hfq (Fig. 5b). On northern blot, FadZ accumulated as a very short species of only ~40 nt in *Salmonella* and in *E. coli* (Fig. 5e, Supplementary Fig. 8). As expected for an Hfq-associated sRNA, the expression of FadZ was abolished in a *Salmonella hfq*-deletion strain, but unaffected in the strain lacking the second global RNA chaperone ProQ (Supplementary Fig. 8c).

FadZ becomes undetectable at the non-permissive temperature (44 °C) in an RNase E temperature-sensitive strain (*rne*^TS) (Fig. 5f), suggesting that it is a processed, 3'UTR-derived sRNA. Consistent with this result, we noticed that the 5' sequence of FadZ matches the consensus motif for RNase E cleavage (Fig. 5d)[25]. Mutating three conserved uridines indeed disrupted cleavage and production of FadZ (Supplementary Fig. 8e), confirming that FadZ is a 3'UTR-sRNA processed by RNase E. Altogether, these data demonstrate that FadZ is a short, Hfq-dependent sRNA cleaved off from the conserved region of the *fadBA* 3'UTR.

## FadZ represses several major porin mRNAs by direct base-pairing

Gene regulation by sRNAs is often mediated via imperfect base pairing with target mRNAs. Consistent with iRIL-seq data (Fig. 4b), RNAhybrid predicts that FadZ base-pairs to the 5' CDS region of the *ompD* mRNA (Fig. 5g). Interestingly, this CDS region is conserved in several other porin mRNAs including *ompC*, *ompN* and *ompS*, indicating FadZ may regulate multiple porins. Indeed, SDS-PAGE analysis of total proteins

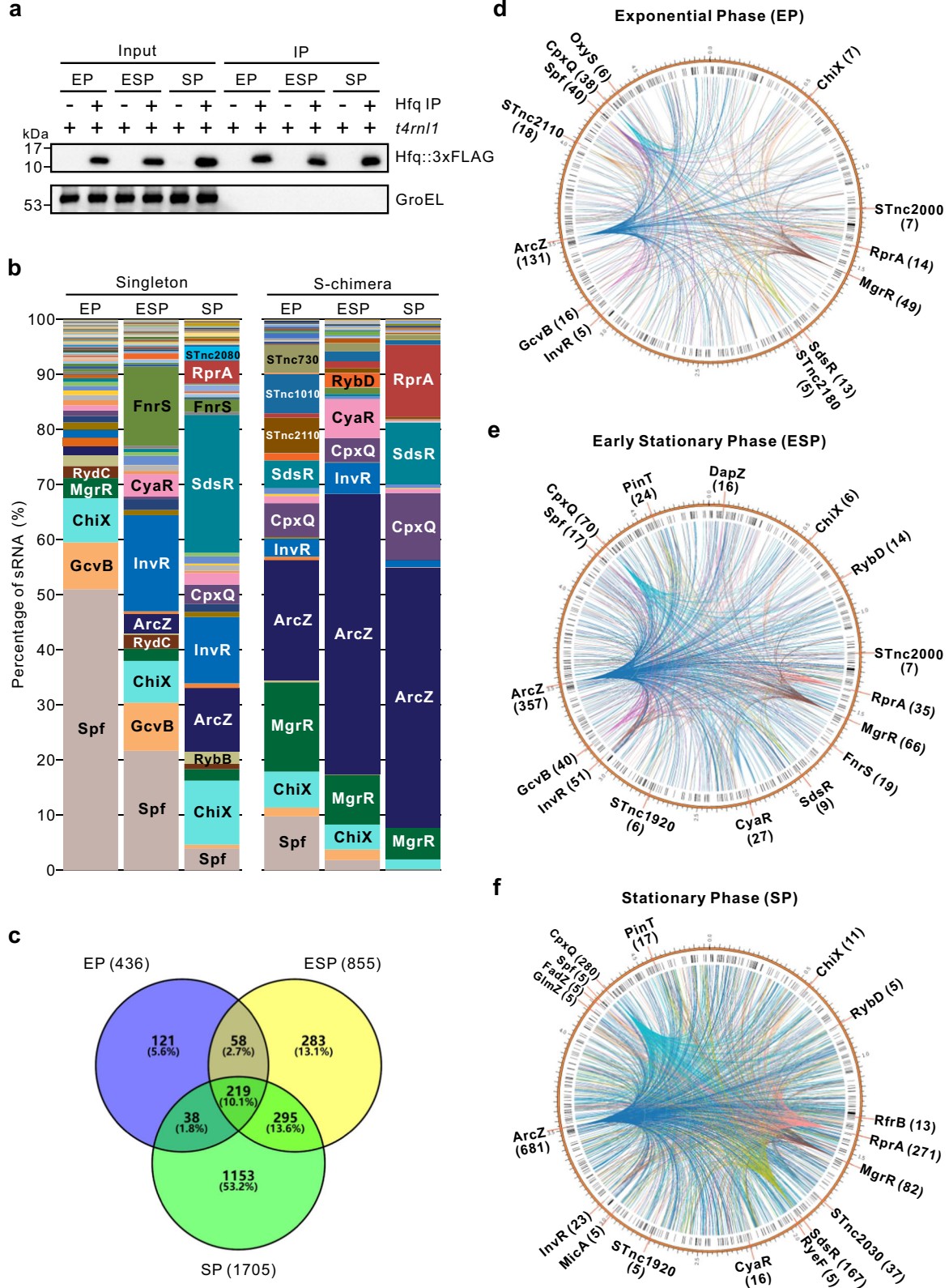

showed that the abundant porins OmpD and OmpC were repressed by constitutive expression of FadZ, whereas a FadZ-M variant with a single point mutation failed to regulate both porins (Fig. 5h, Supplementary Fig. 8a). This result was further corroborated using the standard two-plasmid system and translational-sfGFP reporters[53]. FadZ repressed the expression of all four porins (OmpC/D/N/S) at the post-transcriptional level (Fig. 5i). The FadZ-M mutant abolished the regulation of sfGFP fusions to OmpC and OmpD (Fig. 5j, k). Introduction of a compensatory G to C mutation in the target mRNAs (Fig. 5g) finally restored the regulation by FadZ-M. Altogether, these data conclude that FadZ targets a conserved coding region in porins mRNAs and represses the expression of multiple porins via direct base-pairing.

**Fig. 2 | Global profiling of RNA-RNA interactome by iRIL-seq across growth stages. a** Western blot confirmed the pulldown of Hfq at three growth stages. *Salmonella* WT and Hfq::FLAG strains carrying pYC582 were grown in LB. T4 RNA ligase was induced with L-arabinose for 30 min, before reaching the indicated growth stages (EP: Exponential Phase at OD 0.5, ESP: Early Stationary Phase at OD 2.0, SP: Stationary Phase at OD 2.0 + 3 h). Cells were collected and subjected to iRIL-seq. Input, total proteins from bacterial lysates. IP, proteins after immunoprecipitation using an anti-FLAG antibody. Blot shown is representative of *n* = 2 biological replicates. **b** Relative fraction of individual sRNAs as singleton and S-chimeric fragments in iRIL samples. Percentage represents the reads of a given sRNA compared to all reads from top 100 sRNAs in a library. **c** Venn diagram analysis of RNA–RNA interactions at three growth stages. **d**–**f** Circos plots represent RNA interaction networks at three growth stages. RNA-RNA interactions from three growth stages were represented on the *Salmonella* chromosome. Circumference: Interacting sRNAs in order of genomic context. Labeled sRNAs interact with at least five putative targets, and their interactions are shown in color. Other interactions are black. Source data are provided as a Source Data file.

## FadZ is part of an incoherent feed-forward loop in fatty acid metabolism

To investigate the physiological function of FadZ sRNA, we sought to elucidate upstream signals that activate FadZ expression. Because its parental mRNA *fadBA* is repressed by the transcriptional regulator FadR and derepressed by the addition of certain fatty acids[54], the 3′UTR-derived FadZ may be under similar transcriptional control. Indeed, FadZ levels were elevated in a mutant lacking the *fadR* gene (Fig. 6a). While FadZ was not detectable in minimal medium containing glucose as the sole carbon source, it was strongly induced upon the supplementation with long-chain (Ole, oleic acid, C18:1) as well as medium-chain (Oct, octanoic acid, C8:1) fatty acids. These data confirm that FadZ, as well as its parental *fadBA* mRNA, is activated by the availability of fatty acids. Under this condition, expression of OmpD and OmpC are completely repressed by overexpressing FadZ (Fig. 6b), suggesting that the sRNA may function to shut down the expression of these abundant porins during fatty acid metabolism.

Additional transcription factors may control the expression of *fadBA* and FadZ, since FadZ accumulates in *Salmonella* when growing in LB. Screening a small panel of regulators identified CRP as an upstream activator of FadZ expression (Fig. 6c, d). FadZ expression and levels of a *fadBA-lacZ* transcriptional reporter were extremely low in a Δ*crp* mutant. CRP has also been suggested to regulate *ompD* expression[55]. Using a transcriptional *ompD-lacZ* reporter fusion, we indeed confirmed that CRP activates transcription from the *ompD* promoter (Fig. 6e, Supplementary Fig. 8b). These findings support that *ompD* and its sRNA repressor FadZ are activated by the same upstream transcriptional regulator CRP, thus forming a type-1 incoherent feed-forward loop (Fig. 6f). Finally, we have observed an obvious growth defect when FadZ was constitutively expressed in medium containing oleic acid, and an even more pronounced detect for the Δ*crp* mutant (Fig. 6g, h, Supplementary Fig. 8f), indicating a crucial role of this feed-forward loop in fatty acid metabolism.

## Discussion

The RNA-RNA interactome is the core layer in the post-transcriptional control of global gene expression. In this study, we report iRIL-seq as an in vivo approach to profile RNA-RNA interactomes in live cells, by combining in vivo RNA proximity ligation and immunoprecipitation, without crosslinking or challenging in vitro steps such as nuclease trimming. Harnessing this approach, we have established the first global RNA interaction network across multiple growth stages for *Salmonella*, and discovered a number of mRNA regulatory hubs that are targeted by multiple sRNAs. Our characterization indicated that the large, OmpD-centered hub is regulated by twelve sRNAs (Fig. 6i), including a novel 3′UTR-processed sRNA FadZ in the fatty acid metabolism. These mRNA hubs and their sRNA regulators highlight the rapidly growing complexity of RNA regulatory networks in prokaryotes and their vital importance in bacterial physiology.

### iRIL-seq is a reliable and straightforward in vivo approach

We utilized iRIL-seq to profile the global RNA interactome in *Salmonella* at three growth stages. In total, we have identified ~2100 unique and significant interactions involving 128 sRNA candidates and 1122 mRNAs, which represents Hfq-mediated regulation of up to a quarter of all *Salmonella* genes. Therefore, iRIL-seq provides a comprehensive in vivo view of the post-transcriptional regulatory network in a model bacterial pathogen.

iRIL-seq has several advantages over other RNA interactome mapping approaches. A significant advantage is the ability to identify physiologically important interactions in vivo, and to identify the relevant base-pairing regions in the S-chimeras. The natural ends of interacting RNAs are ligated on Hfq in situ and enriched by Hfq-coIP under native conditions. Because iRIL-seq does not require UV cross-linking and RNase trimming prior to ligation in test tubes[23,24], we speculate that iRIL-seq may generate fewer transient interactions that are difficult to interpret and validate[56]. Importantly, iRIL-seq successfully identified complementary sequence motifs in S-chimeras for ~95% of the predicted target mRNAs with extraordinary confidence (Fig. 3c–f, Supplementary Fig. 6), demonstrating that the ligations reliably occur between base-paired RNAs on Hfq in vivo.

Another advantage of iRIL-seq is its simpler setup under in vivo conditions, with equally rich information that is obtained by other methods. Similar to Hi-GRIL-seq approaches[22,32], the setup of iRIL-seq requires only a plasmid to express T4 RNA ligase in vivo, alleviating the need of expensive RNA-grade enzymes for in vitro reactions[37]. iRIL-seq is highly streamlined and may be performed within a single day by students with minimal training/experience in RNA biochemistry. Our in vivo approach requires less amount of input cells, potentially allowing future studies in infected host cells or at the single-cell level. Due to the complex nature of RIL-seq and CLASH protocols, the RNA-interactome studies have so far limited to few RNA-binding proteins in model pathogens under lab conditions[40]. The broader application of iRIL-seq may help decipher the dynamic RNA-RNA interactome associated with different RNA binding-proteins under various environmental stresses and infection conditions. We envision that iRIL-seq may become a prime choice for dissecting the RNA-interactome in hundreds of other non-model and model bacterial organisms, in which the Hfq regulatory network is still waiting to be explored[57,58].

It is important to note that in vivo ligation may also have limitations. For example, the expression of T4 RNA ligase needs to be activated by an inducer (e.g., L-arabinose), which may affect the transcriptome during induction. Besides Hfq, the iRIL-seq approach needs further validation for other RNA-binding proteins with varying affinity to sRNAs, such as ProQ and CsrA, where UV crosslinking might become necessary to stabilize RNA chimeras. Future addition of end-repairing steps in vivo (e.g., RppH) might improve ligation with primary sRNAs by iRIL-seq.

### iRIL-seq enables identification of processed sRNAs and their targets

An increasing number of studies report novel regulatory sRNAs that are processed from various regions inside mRNAs or tRNAs[2,3,19,21,59]. Despite the growing interest in the class of processed sRNAs, there is a lack of specialized method to globally identify and differentiate them from degradation intermediates. The discovery of processed sRNAs has been previously aided by mapping the RNase E-cleavage fragments using TIER-seq, and by identification of Hfq-associated RNAs using RIP-seq in our earlier studies[20,25]. Building on Hfq RIP-seq[20], the iRIL-seq

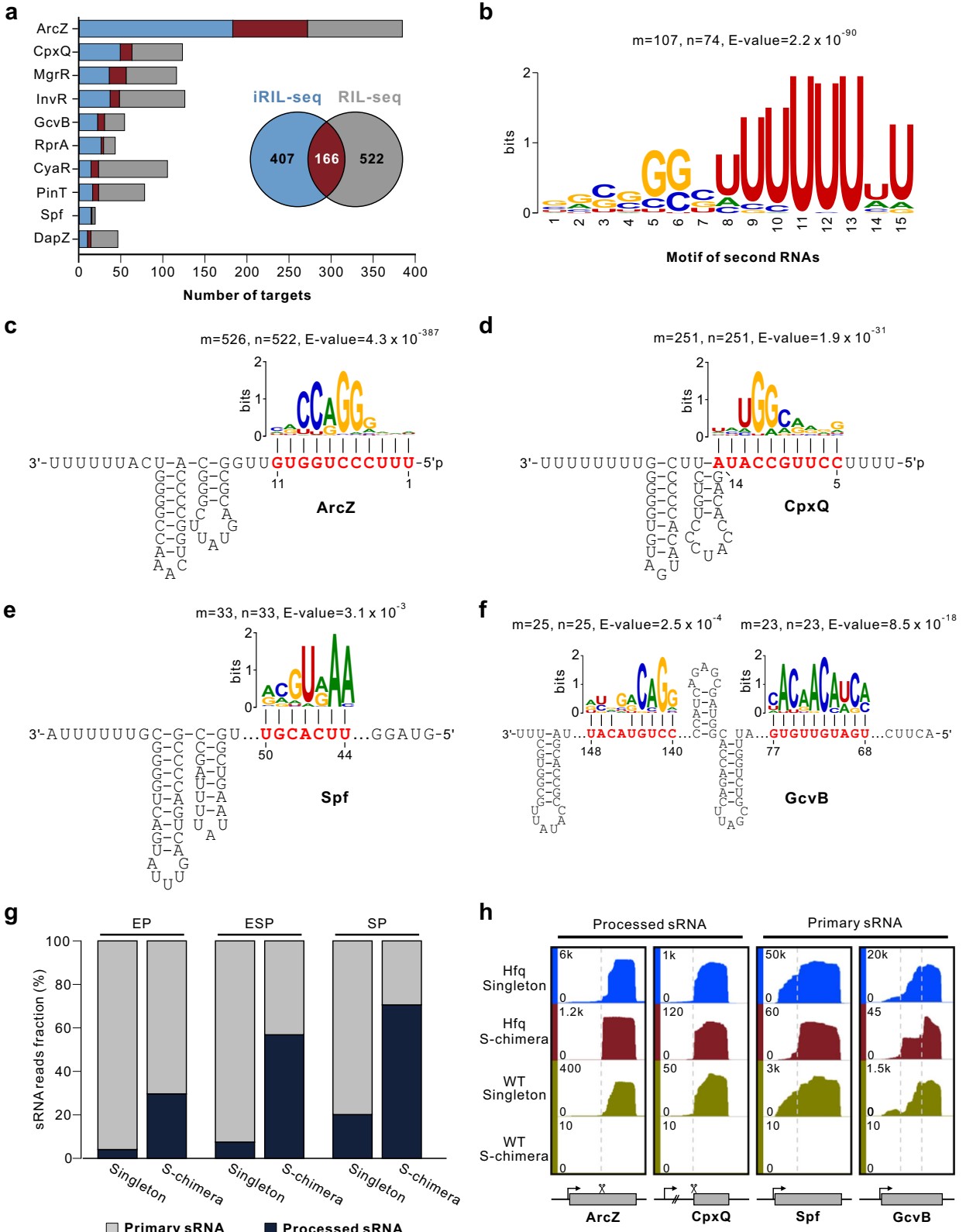

**Fig. 3 | Characterization of sRNA-target interactions in iRIL-seq datasets.**
**a** Comparison of the sRNA-mRNA interactions found by iRIL-seq and RIL-seq under ESP condition. The bars indicate the numbers of predicted targets for 10 sRNAs that have most targets predicted. Venn diagram (inset) shows the overlap of all predicted targets for these 10 sRNAs between the two datasets. **b** Sequence motif identified in RNA2 in S-chimeras. M indicates the total number of RNA2 sequences. N indicates the number of RNA2 sequences containing the motif. **c–f** Motifs identified in the targets are complementary to the cognate sRNAs. m, the total number

of target sequences. n, the number of target sequences containing the motif.
**g** Known processed sRNAs (Supplementary Data 6) were enriched in S-chimeras compared to primary sRNAs. Similar trend was also observed by RIL-seq (Supplementary Fig. 3). **h** Genome browser screenshots showing the genomic locations of sRNAs as singleton and S-chimeric fragments in the iRIL-seq dataset at ESP. Dashed lines indicate the 5′ end of sRNA fragments found in S-chimeras in Hfq-associated iRIL-seq data.

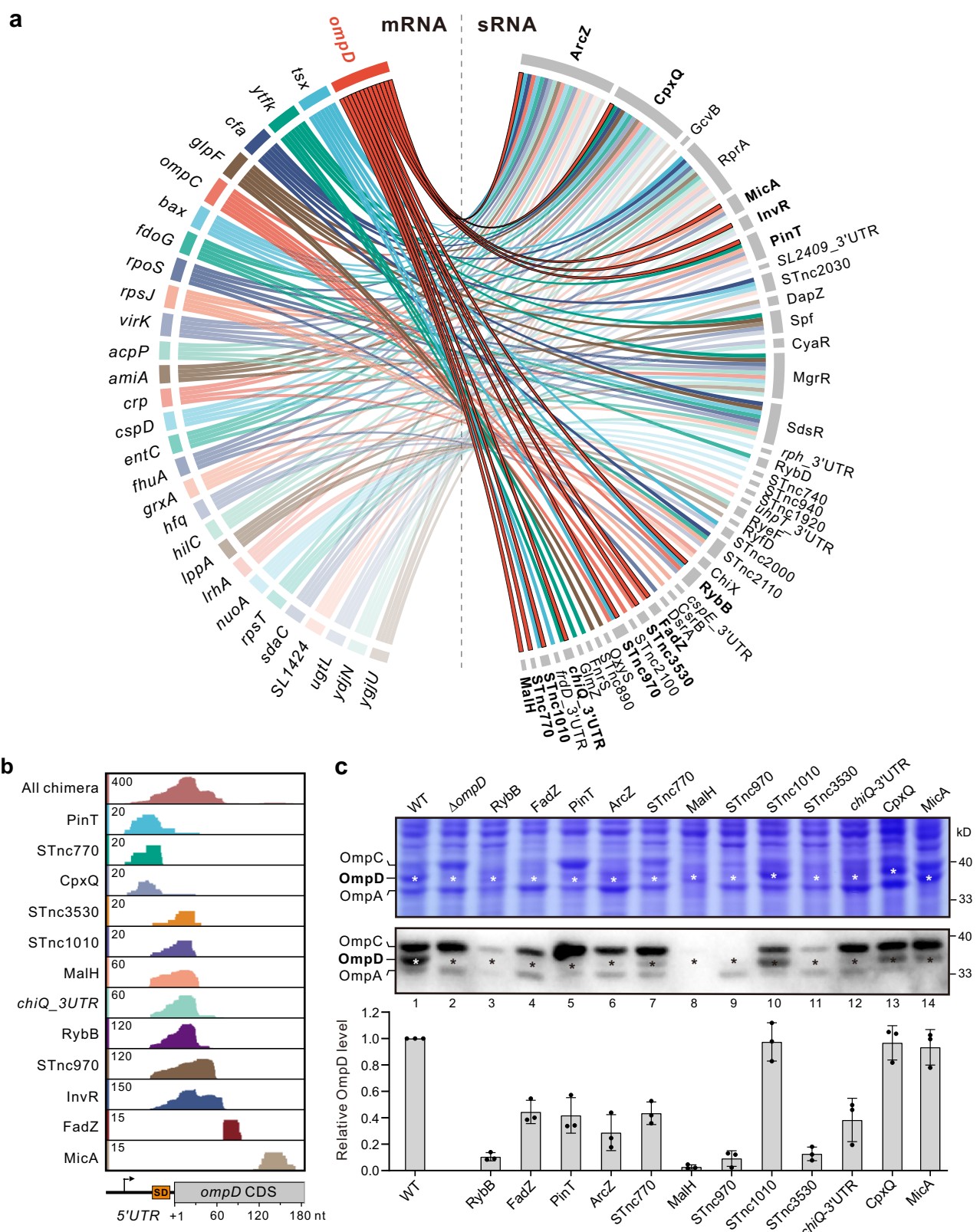

**Fig. 4 | iRIL-seq determines mRNA regulatory hubs. a** Circos plot showing the mRNA regulatory hubs that interact with at least four sRNA candidates. The hub mRNAs were depicted on the left hemisphere, with their interacting sRNAs on the right. sRNAs interacting with *ompD* are in bold. **b** Genome browser screenshots showing chimeric reads mapped to the 5' region of *ompD*. The location of Shine-Dalgarno sequence was indicated as an orange box. **c** Verification of OmpD regulation by different sRNAs. The *Salmonella* WT strain containing an empty vector or sRNA overexpression plasmids was grown overnight in LB. Total proteins were analyzed by 12% SDS-PAGE. The gel was stained with Coomassie brilliant blue, or subjected to Western blotting using a polyclonal anti-OMP antiserum. Δ*ompD* served as a OmpD-null control. The OmpD bands are indicated by asterisks. The levels of OmpD from three replicates are quantified. Blot shown is representative of *n* = 3 biological replicates. Graph bar represents mean relative protein abundance. Error bars indicate standard deviations from *n* = 3 biological replicates. Source data are provided as a Source Data file.

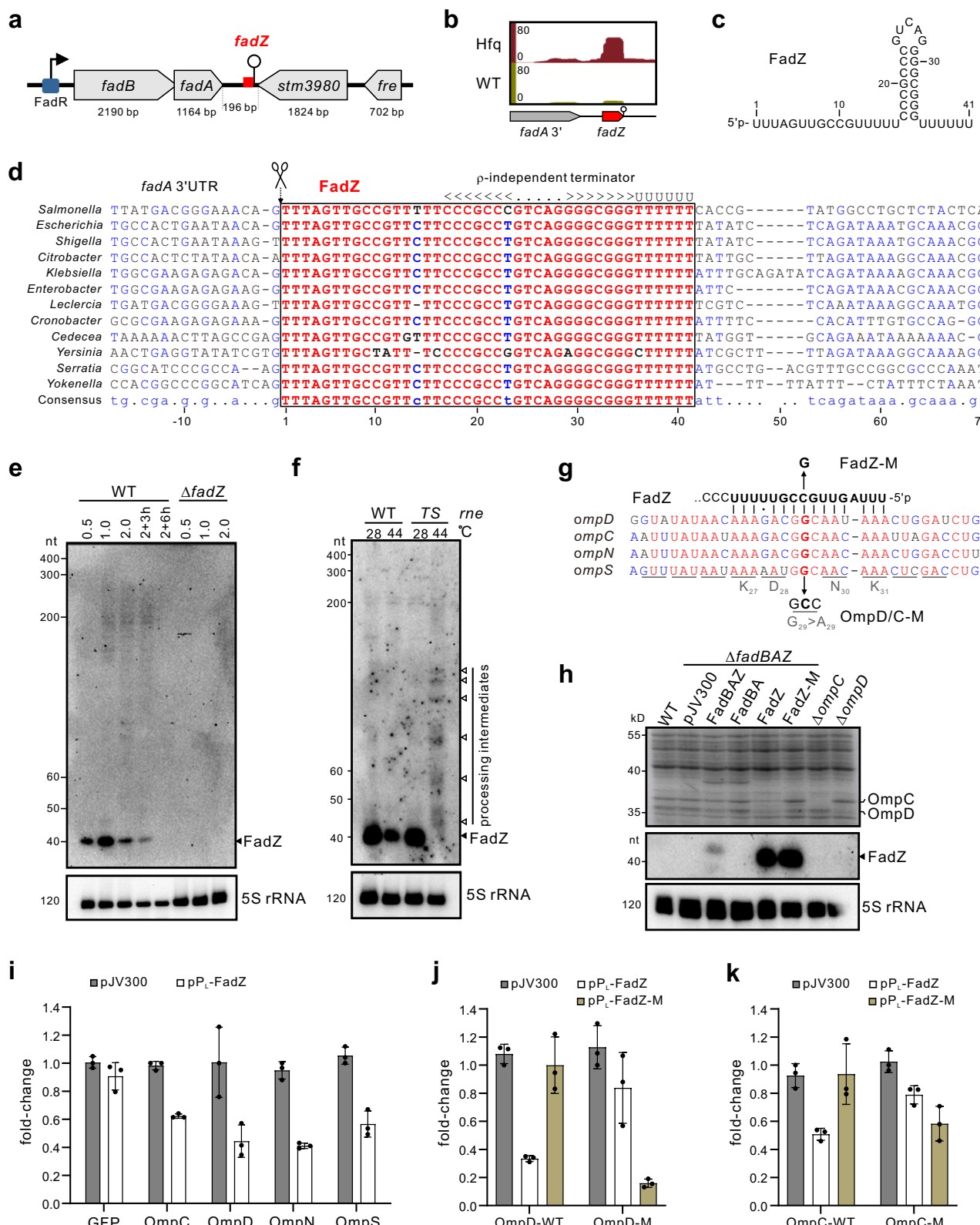

approach here represents an effective method to differentiate processed, functional sRNA regulators from degradation intermediates. Our results show that iRIL-seq readily captures processed sRNAs and their abundant ligation products in chimeric reads (Fig. 3), facilitating both identification and functional assignment in a single experiment. For example, several among those OmpD regulating sRNAs in this study are processed fragments, including FadZ, MalH (STnc810), STnc970, and STnc1010, all of which are derived from the 3'UTR of mRNAs[25,60].

Since sRNAs are often at the 3' end of S-chimeras (Fig. 1f, Supplementary Fig. 4), searching RNA2 sequences for mRNA-derived fragments could discover novel processed sRNAs. Indeed, we have found 3'UTRs of 18 protein-coding genes located in RNA2 in our iRIL-seq datasets (Supplementary Data 7). These are likely Hfq-associated

**Fig. 5 | FadZ is a 3'UTR-processed sRNA repressor of the OmpD porin.**
**a** Schematic of the genomic context of the *fadBA* operon and FadZ in *Salmonella*.
**b** FadZ was enriched by Hfq-coIP. Genome browser screenshot showing the
genomic locations of FadZ fragments under ESP condition. **c** The sequence and
predicted secondary structure of FadZ. **d** Alignment of FadZ genomic sequences
from representative enterobacterial genera. Conserved nucleotides were marked in
red. The processed 5′ end and Rho-independent terminator were indicated.
**e** Expression of FadZ in *Salmonella* wild-type and FadZ-deletion strains. Total RNA
was isolated at indicated time points and analyzed on northern blot. 5S rRNA was
probed as a loading control. **f** Northern blot analysis of FadZ expression in a strain
carrying a temperature-sensitive RNase E allele (*rne*-TS) and the wild-type allele
(*rne*-WT). Bacteria were grown in LB-broth to OD 1.0 at 28 °C and then shifted to
28 °C or 44 °C for 30 min. Total RNA was extracted and separated on a 6% PAA gel.

**g** Predicted binding sites between FadZ and OMP genes using RNAhybrid. Con-
served nucleotides among OMP genes were marked in red. Amino acids corre-
sponding to the binding sites were shown below. The mutated nucleotides were
indicated. **h** Regulation of abundant major porins by FadZ. *Salmonella* strains
containing control vector pJV300 or plasmids constitutively expressing the indi-
cated RNA fragments were grown overnight in LB. Total proteins were analyzed by
SDS-PAGE, and total RNAs were analyzed by northern blotting. **i** Relative fluores-
cence for translational OMP-sfGFP reporters. A Δ*fadZ* strain carrying either the
control plasmid pXG-1 or pXG10-sfGFP with an in-frame fusion to the indicated
genes, together with pJV300 or pPL-FadZ as indicated. **j** Regulation of *ompD*::GFP
by FadZ. **k** Regulation of *ompC*::sfGFP by FadZ. Graph bars (**i–k**) represent mean
relative fluorescence. Error bars indicate standard deviations from $n = 3$ biological
replicates. Source data are provided as a Source Data file.

3'UTR-derived sRNAs, based on the presence of RNase E cleavage sites
and Rho-independent terminators. Several of them are recently char-
acterized as bona-fide sRNAs, such as the *uhpT* 3'UTR[61,62], and the *ahpF*
3'UTR[33]. In other words, iRIL-seq may be an ideal approach to discover
processed intragenic sRNAs and immediately identify their targets,
enabling the elucidation of mRNA-mRNA crosstalk networks in
bacteria.

## Target-centric analysis reveals mRNA regulatory hubs
When we shifted focus from sRNA to mRNAs, our target-oriented
analysis discovered dozens of genes that are targeted by more than
three sRNAs, conceptualized as mRNA regulatory hubs. The major
porin OmpD, as one of the most abundant proteins in *Salmonella*, was
identified in our dataset as one of the largest mRNA regulatory hubs
that was predicted to interact with >10 sRNAs (Fig. 4a, b). We con-
firmed that most of these sRNAs inhibit the expression of OmpD, as
predicted by iRIL-seq (Fig. 4c). The discovery of mRNA regulatory hubs
such as OmpD highlights the large redundancy in sRNA-mediated
regulation. Phenotypes and dysregulation of target genes are not
always visible upon deletion of a single sRNA. Besides FadZ, OmpD is
repressed by four other well-characterized sRNAs (Fig. 6i), including
the *Salmonella* virulence-related sRNA InvR[49], MicC[63], RybB[7], and
SdsR[64]. Not surprisingly, global inactivation of these sRNAs in an *hfq*
mutant led to a strong increase in OmpD levels, resulting in con-
stitutive induction of envelope stress[65].

Apart from *ompD*, we detected ~30 hub genes that interact with at
least four sRNAs. These hubs include mRNAs encoding another
abundant porin (OmpC), RpoS, CRP, Cfa, as well as Hfq itself, most of
which play crucial functional or regulatory roles in bacteria. For the
general stress factor RpoS, our data suggest it may interact with its
downstream regulator SdsR[64], as well as MgrR in the PhoPQ
pathway[66,67], in addition to the three known activating sRNAs (ArcZ,
RprA, DsrA). This suggests that the even well-studied hub genes such
as RpoS may have additional novel sRNA regulators that remain to be
discovered using RNA-RNA interactome approaches. Of note, we have
not detected chimeras for the CsgD hub and its targeting sRNAs[8],
probably because this master regulator of biofilm formation is not
expressed under the conditions used in this study. This indicates that
many more key regulatory hubs may be waiting to be discovered in
bacteria under different growth conditions.

## FadZ is part of a feed-forward loop in response to fatty acid
Fatty acids are essential components of cell wall and are important
sources of metabolic energy for enterobacteria in human gut[68]. Long-
chain fatty acids are taken up by a high-affinity porin FadL, and
metabolized by pairwise removal of carbon atoms through successive
rounds of β-oxidation, releasing ATP and other molecules including
acyl-CoA, acetyl-CoA, NADH, and FADH[69]. This makes fatty acid oxi-
dation crucial for many other catabolic processes such as the TCA and
glyoxylate cycles, as well as the synthesis of cell wall components. The
fatty acid metabolism pathway is known to be regulated by the master

transcription factor FadR[54]. Downstream of FadR, we have found FadZ
as the missing cognate sRNA regulator in this pathway in enter-
obacteria. FadZ is processed by RNase E from the 3'UTR of *fadBA*
mRNA, and represses major porins OmpD and OmpC (Fig. 6i). Because
the abundant OmpC/D porins have low affinity to fatty acids, their
reduction may remodel the bacterial membrane and provide more
room for FadL, an alternative porin that imports medium/long-chain
fatty acids with high efficiency[70]. Intriguingly, a recent RIP-seq study in
*V. cholerae* has identified another 3'UTR-sRNA FarS that is processed
from a different mRNA encoding fatty acid biosynthesis genes[14]. FadZ
and FarS share no sequence homology or apparent common evolu-
tionary origin. Activated by FadR, *Vibrio* FarS actives the expression of
fatty acid biosynthesis gene *fabB* and represses oxidation gene *fadE* at
the post-transcriptional level, together forming a type 3 coherent feed
forward loop. Thus, two different sRNAs have independently evolved
from mRNA 3'UTRs in two distant organisms to facilitate post-
transcriptional regulation in response to fatty acid metabolism.

The biogenesis of FadZ is under multiple layers of control in *Sal-
monella* (Fig. 6i). Its parental mRNA *fadBA* is activated by CRP and
repressed by FadR at the transcriptional level, while FadZ processing is
activated by RNase E. Mature FadZ binds to a conserved region deep in
the CDS of several major porins mRNAs, resulting in translational
repression likely by promoting mRNA decay[63]. It is unclear whether the
5′ monophosphate of FadZ contributes to the recruitment of RNase E[71].
We have identified FadZ as part of type I incoherent feed forward loop
(I1-FFL). This type of regulatory loop is not uncommon for sRNAs in
bacteria, for example PhoP-AmgR-*mgtC* in *Salmonella*[72], PrrA-PcrZ-
*bchN* in Rhodobacter[73], and involving a 3'UTR-derived sRNA, NarL-
NarS-*nirC* in *Salmonella*[30]. The results here add FadZ as another
example of 3'UTR-derived sRNA mediated I1-FFL. In this case, the sRNA
might function as a fine-tuning mechanism to minimize fluctuation in
target mRNA levels in response to fatty acid oxidation.

## Methods
### Bacterial strains and growth conditions
Bacterial strains used in the study can be found in the Supplementary
Data 1. *Salmonella enterica* serovar Typhimurium strain SL1344 was
used as wild-type. Strains with deletions or chromosomally 3xFLAG
epitope-tagging were constructed using the λ-Red recombinase
method[74,75]. Bacteria were grown at 37°C with 220 rpm shaking in
lennox broth (LB) or M9CA minimal medium (#A507025) supple-
mented with 0.1% carbon source: glucose (#A501991), octanoic acid
(#A501844), oleic acid (#A502071) at pH 7.0. Brij 58 (#A606306) was
added at a final concentration of 0.4% as vehicle to promote dissolu-
tion of fatty acids. For T4 RNA ligase pulse expression, bacteria were
grown in LB at 37 °C to the indicated $OD_{600}$, L-arabinose was then
added to the final concentration of 0.2% for 30 min. For the *rne*-TS
experiment, bacteria were grown in LB at 28 °C to an $OD_{600}$ of 1.0, and
then shifted to 28 °C or 44 °C for 30 min. For oleic acid exposure
assays, overnight cultures were grown from a single colony in M9CA
minimal medium supplemented in 0.1% glucose with 0.4% Brij-58,

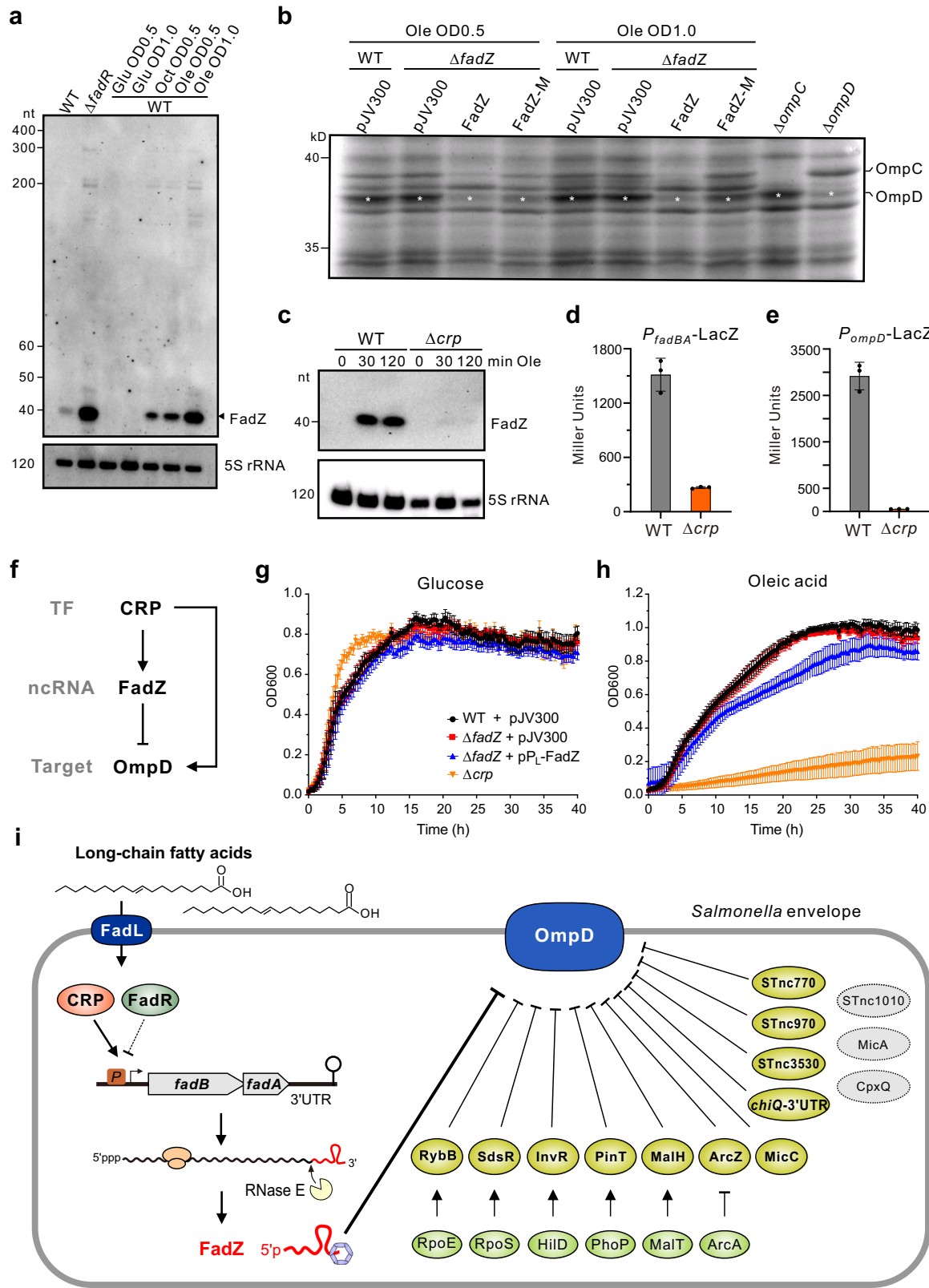

diluted 1:100 in fresh medium and grown to $OD_{600}$ of 1.0. Cells were washed twice with PBS at room temperature and resuspended in M9CA minimal medium supplemented with 0.1% oleic acid and 0.4% Brij-58. Where appropriate, media were supplemented with antibiotics at the following concentrations: ampicillin (Amp), 100 μg/ml; kanamycin (Kan), 50 μg/ml; chloramphenicol (Cm), 20 μg/ml; and hygromycin (Hyg), 100 μg/ml. Unless stated otherwise, chemicals were

purchased from Sangon Biotech, Shanghai. The plasmids were documented in Supplementary Data 2, and oligonucleotides were listed in Supplementary Data 3.

### iRIL-seq experiments

Wild-type *Salmonella* and *hfq*::3xFLAG tagged strains carrying the plasmid pYC582 were grown in LB medium at 37 °C with 220 rpm

**Fig. 6 | FadZ is induced by fatty acid metabolism as part of feed-forward regulatory loop. a** Expression of FadZ in *Salmonella* during growth in M9CA minimal medium supplemented with indicated carbon sources (final concentration of 0.1%). Glu, D-glucose; Oct, octanoic acid (C8); Ole, oleic acid (C18:1). **b** Regulation of major porins by FadZ in the presence of fatty acids. *Salmonella* strains containing control vector pJV300 or pZE12-FadZ were grown in M9CA minimal medium supplemented with 0.1% oleic acid. Total proteins were analyzed by SDS-PAGE. The OmpD bands are indicated by asterisks. **c** Expression of FadZ in wild-type and Δ*crp* during oleic acid shock at indicated time points. **d**, **e** β-galactosidase activities of a chromosomally encoded *lacZ*-transcriptional fusions to indicated promoters. Cells were grown in M9CA medium containing 0.1% glucose to an $OD_{600}$ of 1.0, and then resuspended in M9CA medium containing 0.1% oleic acid for 1 h before assay. Error bars indicate standard deviations ($n = 3$). Graph bars (**d**, **e**) represent average activities. Error bars indicate standard deviations from $n = 3$ biological replicates. **f** FadZ-mediated type I incoherent feed-forward loop (I1-FFL). TF, transcription

factor upstream. Arrow refers to activation and bar refers to repression. Growth curve of *Salmonella* strains in M9CA medium containing 0.1% glucose (**g**) or 0.1% oleic acid (**h**). **i** Model showing that the 3′UTR-derived FadZ sRNA regulates a central regulatory hub OmpD. *Salmonella* major porin OmpD is a key mRNA regulatory hub regulated by twelve different sRNAs, which are under different transcriptional control. Dashed ovals indicate three sRNAs that potentially interact with *ompD* but showing no regulation. FadZ is the cognate sRNA in the fatty acid metabolism pathway. Extracellular long-/medium-chain fatty acids are transported into bacteria by a specialized porin FadL, and activate the expression of the *fadBAZ* mRNA via two master transcriptional regulators, CRP and FadR. The 3′UTR of mRNA is cleaved by RNase E to produce the FadZ sRNA. With the help of Hfq, FadZ basepairs to the porin mRNAs to repress their expression and trafficking into the *Salmonella* envelope. Data points (**g**, **h**) represent average $OD_{600}$ value. Error bars indicate standard deviations form $n = 3$ biological replicates. Source data are provided as a Source Data file.

shaking. T4 RNA ligase was induced for 30 min by addition of 0.2% L-arabinose when the culture reached the indicated $OD_{600}$ (0.3, 1.5, 2.0 + 2.5 h). The strains were grown to EP (exponential phase, OD 0.5), ESP (early stationary phase, OD 2.0) and SP (stationary phase, OD 2.0 + 3 h), respectively. A culture volume corresponding to 50 $OD_{600}$ (e.g., 100 ml for cells at $OD_{600}$ of 0.5) of bacterial cells were collected by centrifugation at 12,000 g for 5 min. The pellets were washed twice with 10 ml precooled PBS and stored at −80 °C until use. Hfq-coIP was performed using a published protocol in our previous reports[20]. The bacterial pellets were resuspended in 600 µl ice-cold lysis buffer (20 mM Tris pH 8.0, 150 mM KCl, 1 mM of $MgCl_2$, 1 mM DTT, 0.05% Tween-20), and lysed for 10 min with 500 µl glass beads using Cryolys Evolution at 4 °C. Cell lysates were collected by centrifugation for 30 min at 17,000 g at 4 °C, and transferred to new tubes. The lysates were incubated with anti-FLAG antibody (#F1804, Sigma-Aldrich) conjugated protein G magnetic beads (#10004D, Thermo Fisher) for 1 h at 4 °C with rotation. The beads were washed five times with 500 µl ice-cold lysis buffer and resuspended in 100 µl lysis buffer. The Hfq-bound RNA was purified using RNA clean & concentrator columns (#B518688, Sangon) and resuspended in nuclease-free water.

## Library construction and RNA-seq
iRIL-seq libraries were constructed using the RNAtag-Seq protocol with a few modifications[37]. Briefly, RNA was subjected to fragmentation and DNase I treatment, and purified with 2.5x Agencourt RNAclean XP beads (#A63987, Beckman-Coulter) and 1.5x isopropanol. RNA was ligated to the 3′ barcoded adaptor and purified with 2.5x AMPure XP beads and 1.5x isopropanol. Ribosomal RNA was removed using Ribo-off rRNA depletion Kit (#N407-01, Vazyme). The rRNA depleted sample was purified with 2.5x RNAclean XP beads and 1.5x isopropanol. First strand cDNA was synthesized using HiScript II 1st Strand cDNA Synthesis Kit (#R211-01, Vazyme). RNA was degraded by 1 M NaOH. cDNA was purified with 2.5x AMPure XP beads and 1.5x isopropanol. The cDNA 3′ end was attached to a second adaptor and cleaned up twice with 2.5x RNAclean XP beads and 1.5x isopropanol. Libraries were PCR amplified with Illumina P5 and P7 primers using Q5 High-Fidelity DNA Polymerase (#M0491L, NEB), and purified with 1.5x RNAclean XP beads. The libraries were sequenced by 150 bp paired-end sequencing with an Illumina NovaSeq 6000 instrument.

## RNA-seq data analysis
The sequencing data were analyzed similar as previous RIL-seq methods[23,37]. The raw sequencing reads of iRIL-seq in fastq files were processed by cutadapt to trim the adaptor sequence and remove low-quality ends. In order to detect chimeric fragments, the first 25 nucleotides of high-quality paired-end reads were taken and mapped to the genome of *Salmonella* strain SL1344 (NC_016810.1) using the BWA software with default parameters. The paired reads obtained

from paired-end sequencing by Illumina NovaSeq are considered two mates. If two mates of 25 nt from both ends of a sequenced fragment were mapped to different genomic locations and the distance is greater than 1000 nt, the fragment was called as chimera. Otherwise, the fragment was defined as singleton when the mates mapped to the same transcript or within a distance of 1000 nt. Significant chimeras (S-chimera) were analyzed by comparing the detected pairwise interactions to the counts of random ligations using a Fisher's exact test as previously reported[23,37]. Each Chimera was assigned a *p*-value and an Odds Ratio value by multiple hypotheses testing. Chimeras with a *p*-value ≤ 0.05 and an Odds Ratio ≥1 were defined as S-chimeras. Only S-chimeras with ≥10 sequenced fragments were presented and further investigated. For the genome annotation of RNA fragments, the genomic features were classified into eight major categories: housekeeping RNA (hkRNA: RnpB, SsrS, Ffs and SsrA), tRNA, rRNA, CDS, IGR (intergenic region), sRNA, 3′UTR, CDS, 5′UTR.

## Analysis of sequence features in RNA2 and sRNA targets
To analyze the sequence features of RNAs located in RNA2, all the sequences of RNA2s in Supplementary Data 5 were extracted. For multiple RNA2 duplicates, only RNA2 containing the longest sequence was selected. The consensus motif was identified using the MEME suite (v5.5.2) with default parameters[76]. The consensus motifs related to Supplementary Fig. 5 was generated using the RNA2 sequences derived from three growth stages. 107 unique RNA2 sequences were extracted and identified the consensus motif related to Fig. 3b. To search the common motif of sRNA target sequence, we selected sRNAs with at least seven different putative targets. Targets sequences of these sRNAs were extracted and analyzed by MEME allowing motif width to range from 6 to 15 nucleotides[76]. Only motifs that had an E-value ≤ 0.05 were considered.

## SDS-PAGE and Western blotting analysis
Bacterial samples were collected and resuspended in 1× protein loading buffer and incubated at 95 °C for 5 min to lyse the cells. For SDS-PAGE, 0.1 OD of total protein samples were loaded per lane. Gels were stained overnight with Coomassie blue and visualized using ChemiDocTM XRS+ (Bio-Rad). For Western blotting, 0.05 OD of proteins samples were loaded per lane and separated by SDS−PAGE. Proteins were transferred onto nitrocellulose membrane (#10600002, GE Healthcare) and blocked with DifcoTM skim milk (#6307915, BD). Membranes were incubated with monoclonal α-FLAG (Sigma-Aldrich #F1804; 1:1,000), α-His (Sigma-Aldrich #SAB4301134; 1:1,000) or α-GroEL (Sigma-Aldrich #G6532; 1:5,000) antibodies, and secondary α-mouse or α-rabbit HRP-linked antibodies (Sigma-Aldrich #A0168 or #A0545; 1:5,000). Chemiluminescence was developed using the Novex™ ECL Chemiluminescent Substrate Reagent Kit (#WP20005, Thermo Fisher Scientific), and then visualized on ChemiDocTM XRS+ and quantified using ImageLabTM Software.

## GFP fluorescence quantification

*Salmonella* strains carrying GFP translational fusions were grown in LB media containing Amp and Cm to an $OD_{600}$ of 0.5. 100 µl of the cultures were collected and washed 3 times with 1x PBS and fixed with 4% paraformaldehyde. GFP fluorescence intensity was quantified by a microwell plate reader or flow cytometry (FACS Calibur, BD Bioscience).

## β-galactosidase assay

100 µL bacterial culture was collected and mixed with 15 µL 0.1% SDS, 30 µL chloroform and 700 µL Z buffer (60 mM Na2HPO4, 40 mM $NaH_2PO_4$, 10 mM KCl, 1 mM $MgSO_4$, pH = 7.0 and supplemented with 2.7 ml of β-mercaptoethanol per liter). Mixtures were vortexed and incubated at 30 °C for 5 min and then added with 200 µL ONPG buffer (4 mg/ml, prepared in Z Buffer) to start the reaction at 30 °C. When the reaction mix became yellow in the optimal range ($OD_{420}$ 0.2–1.0), 500 µl of 1 M $Na_2CO_3$ was added to stop the reaction. Reaction mixture was centrifugated and 200 µL supernatant was collected for OD measuring. $OD_{420}$ and $OD_{550}$ were read on a Bio-Rad benchmark plus microplate reader with the Z-buffer as blank. β-galactosidase activity was calculated in Miller units with the following formula: $1000 \times (OD_{420} - 1.75 \times OD_{550})/(OD_{600} \times time[min] \times 0.1 [ml])$.

## RNA extraction and northern blot analysis

Bacterial total RNA was isolated using the hot phenol method. Briefly, Cells were resuspended in lysozyme, 10% SDS and 3 M sodium acetate (pH 5.2). The cleared lysate was mixed with saturated phenol (pH 4.5) and incubated at 64 °C for 6 min with shaking. After mixing with chloroform and centrifugation in a Phase Lock Gel tube (#WM5-2302820, TIANGEN Biotech), the aqueous phase was collected and mixed the with 30:1 ethanol: sodium acetate (pH 6.5) and precipitated at −80 °C overnight. RNA pellets were washed with 80% ethanol and dissolved in nuclease-free water. The RNA concentration was determined using NanoDrop 2000. 10 µg of total RNA was denatured at 95 °C for 2 min in RNA gel loading buffer II, and separated by gel electrophoresis on 6% polyacrylamide/7 M urea gels in 1x TBE buffer. RNA was transferred to Hybond-N+ membranes (#RPN203S, GE Healthcare) by electroblotting. For agarose gels, 25 µg of total RNA was denatured at 65 °C for 5 min in RNA loading buffer and separated in 1.2% agarose gels containing 1% formaldehyde in 1x MOPS buffer. Gels were stained with ethidium bromide to visualize rRNA, then transferred onto Hybond-N+ membranes (#RPN203S, GE Healthcare) using capillary blotting in 10x SSC buffer overnight. The membranes were cross-linked with UV light (120 mJ/cm$^2$).

Northern blot analysis was performed using the Roche DIG system. Briefly, membranes were prehybridized in DIG Easy Hyb (#11796895001, Roche) for 30 min. The Digoxin-labeled DNA probe (QGO-1580 for probing FadZ, QGO-077 for probing 5S rRNA) was hybridized at 50 °C overnight. The Digoxin-labeled RNA probe (for probing *fadBA*) was hybridized at 68 °C overnight. Membranes were washed three times, for 15-min each, in 5× SSC/0.1% SDS, 1× SSC/0.1% SDS and 0.5x SSC/0.1% SDS buffers at 50 °C for the DNA probe or 68 °C for the RNA probe. After one wash in maleic acid wash buffer for 5 min at 37 °C, and then blocking solution (#11585762001, Roche) for 45 min at 37 °C, membranes were incubated with 75 mU/mL Anti-Digoxigenin-AP (#11093274001, Roche) in blocking solution for 45 min at 37 °C. Membranes were then washed in maleic acid wash buffer twice for 15 min each, and equilibrated with detection buffer. Signals were visualized with CDP-star (#12041677001, Roche) on a ChemiDocTM XRS+ station and quantified using ImageLabTM Software.

## qRT-PCR

Quantitative RT-PCR was performed with the PrimeScript™ RT reagent Kit (#RR047A, Takara Bio). 1 µg total RNA was treated with gDNA eraser and then reverse-transcribed to cDNA with random oligos and PrimeScript RT Enzyme Mix I. cDNA transcribed from 0.025 µg total RNA was used per PCR reaction with TB Green Premix Ex Taq II (#RR820A, Takara Bio). PCR was performed using a Bio-Rad CFX96 Touch Real-Time PCR Detection System. Data were analyzed by the relative quantification (ΔΔCt) method, with the *rfaH* gene as the reference for normalization.

## Reporting summary

Further information on research design is available in the Nature Portfolio Reporting Summary linked to this article.

## Data availability

The data supporting the findings of this study are available from the corresponding authors upon request. The sequencing data have been deposited in the GEO database under No. GSE234792. Source data are provided with this paper.

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

## Acknowledgements

We thank Jörg Vogel, Sahar Melamed, Sarah Svensson and members of the Chao lab for critical reading and comments. We are grateful to Jörg Vogel for providing the anti-OMP antiserum, Stephen Lory for the pKH13 plasmid, Daniel Falush for computational resource, and Andrew Camilli for past support. This work was financially supported by the National Key R&D Program of China (2022YFE0111800 and 2022YFC2303200 to Y.C.), the Natural Science Foundation of China (32270064 to Y.C., 32170179 and 81991532 to C.W., 32300028 to F.L., 32200031 to K.W.), Chinese Academy of Sciences (XDB0570000 and 176002GJHZ2022022MI to Y.C.), Shanghai Municipal Science and Technology Commission (21ZR1471300 and 2019SHZDZX02 to Y.C.), the State Key Laboratory of Oral Diseases (SKLOD2022OF03 to Y.C.), and China Postdoc Fellowship (2022M713256 to F.L.).

## Author contributions

Y.C. and C.W. conceived the study; F.L., Z.C., S.Z., K.W., C.B., C.W. conducted experiments; F.L., K.W. processed RNA-seq data; F.L., Z.C., C.W., Y.C. analyzed data; Y.C., C.W., F.L. wrote the manuscript; Y.C. and C.W. supervised the project and acquired funding.

## Competing interests

The authors declare no competing interests.
