## [Peer Review File · Nature Communications]

In vivo RNA interactome profiling reveals 3'UTR-processed small RNA targeting a central regulatory hubREVIEWER COMMENTS

Reviewer #1 (Remarks to the Author):

Small regulatory RNAs (sRNAs) play a crucial role in post-transcriptional gene regulation in bacteria, and control a wide range of cellular processes, including metabolic pathways, stress responses, quorum sensing, and virulence factor production. The bacterial RNA interactome ensures a rapid response to environmental change via both trans-acting and cis-acting mechanisms that usually involve base pairing with mRNAs to generate regulatory feedback loops and networks.

Until now, the study of the bacterial RNA-RNA interactome has involved laborious indirect methods that involve in vitro cross-linking and RNA ligation. Here, Liu et al report a novel approach that involves the pulse over-expression of RNA ligase to reveal the RNA-RNA interactome within living bacterial cells. Following extensive validation, the authors used the LiRIP-seq approach to investigate the dynamics of sRNA-mRNA interactions during bacterial growth. The authors report the discovery of a previously unknown regulatory role for the FabZ sRNA, which directly influences growth on fatty acids. The LiRIP-seq data also provide functional insights for a range of other functionally-uncharacterized sRNAs.

The paper is likely to be highly cited because it reports the RNA-RNA interactome of the model pathogen *Salmonella enterica* serovar Typhimurium and the role of 30 hub mRNAs that interact with > 4 sRNAs.

Our current understanding of the RNA-RNA interactome of *Salmonella* is based on 20 years of intensive research. Perhaps a more important impact of the Liu et al's study is that LiRIP-seq could be used to comprehensively catalogue sRNA-mRNA interaction networks in a wide range of bacterial species, without the need for decades of research.

The methods and experimental strategies used in the study were of a particularly high standard. The only significant change that I recommend is to the names given to the three stages of growth used to generate data for Figures 2 and 3. Rather than terming these as "OD 0.5", "OD 2.0" and "OD 2.0 + 3h", I suggest that names are used to build on approaches previously defined for *Salmonella enterica* serovar Typhimurium by the Vogel and Hinton laboratories.

The "OD 2.0" condition has been extensively used to define *Salmonella* Typhimurium gene regulation, and is usually referred to as "ESP" (= early stationary phase). The "OD 0.5" and "OD 2.0 + 3h" conditions do not yet have an accepted abbreviation. I suggest that "OD 0.5" is termed "EP" (Exponential Phase), and "OD 2.0 + 3h" is termed "SP" (Stationary Phase) in Figures 2 and 3, and throughout the paper. These abbreviations should be carefully defined in the Materials and Methods (line 492).

Minor corrections

Line 110: Change "artifact" to "artefact".

Line 312: Add details of the media recipe containing octanoic acid. Add product codes for the oleic acid and octanoic acid fatty acids.

Lines 370-371: Rephrase sentence to improve clarity.

Lines 389 - 404: Paragraph should be improved by avoiding the use of words such as "this", "them", "those" and "their" which introduce ambiguity to some sentences. e.g. at Line 394, what does "their" refer to?

Lines 486 onwards (Materials and Methods): more details are required to allow experiments to be repeated. Manufacturers and product codes should be provided for all media components (Lennox broth, M9CA medium etc).

Line 492: Change "Lenox" to "Lennox".

Line 513: explain the phrase "50 OD" in more detail.

Line 549: Explain the term "mates" in more detail.

Table S1: explain "IG::cat" in more detail.

Table S2: Explain the recombinant DNA manipulations in this table in more detail. For example "via XbaI" lacks the detail needed to repeat the plasmid construction.

Figure 1C: the terms EV and T4 need to be defined in the legend.

Figure 1D legend: add a brief explanation of what an "s-chimera" is.

Figure 2C and Figure 3A: Some fonts used in the Venn diagram should be changed from black to white to improve legibility.

Line 950: Add a brief explanation of what the "Hfq-LiRIP" data refers to.

Figure 5A: Check the annotation of the chromosomal region carefully. Both the STM3980 and STM3981 genes should be located between the *fre* gene and the *fadZ* gene?

Figure 6H and Figure S7F: Change "oleic acids" to "oleic acid".

Line 1006: Change "oleic acids" to "oleic acid".

Reviewer #2 (Remarks to the Author):

The authors present a modified version of the RNA-proximity dependant ligation techniques HiGRIL-seq and RIL-seq for profiling sRNA-mRNA interactions associated with Hfq. The major innovation is that library preparation is significantly simplified which is a major advantage as the current protocols – RIL-seq and CLASH have over 100 steps. This should facilitate broader use of the technique for sRNA interaction profiling. The disadvantage is that protocol does not include trimming or RppH treatment. A native 5'P end is required to facilitate ligation and the dataset is biased towards the subset of processed sRNAs that have a native 5'P and 3'OH end closely associated in vivo. Overall the paper is thorough, very well written, and will be a valuable addition to the sRNA tool kit.

Major comments.

1. It would be useful to clarify the bias in the protocol. Lines 225-237 discuss the preference for processed and 3'UTR sRNAs. The title for this section seems misleading given the authors outline a bias against primary sRNAs. Please include a comparison of RIL-seq and LiRIL-seq ligation at native 5' ends. How biased is the protocol to native 5'P ends? Does the mRNA 3' end have a similar bias and are CDS/3'UTR interactions similarly under represented? What is the distribution of interaction distances from the 5' end and 3' end for LiRIL-seq and RIL-seq or CLASH?
2. A more comprehensive comparison to the RIL-seq datasets would convince the reader that the streamlined protocol recovers equivalent information (or identify areas to be aware of differences). Comparisons of metrics like interaction strength, S-chimeras/interaction, and # S-chimeras recovered

would help demonstrate comparable data. This data is available for Salmonella and E. coli. Figure 3A seems to focus on LiRIL-seq recovered interactions rather than comparing the techniques.

3. HiGRIL-seq includes RppH treatment to remove the 5'PPP cap from RNAs to make them ligation compatible. Can the authors comment on whether this is a useful future addition to the protocol (ie: does HiGRIL-seq recover 5'PPP ends?).

4. The RNAs recovered by LiRIL-seq are surprisingly short given at there is no trimming of RNAs (eg: Figure 1B), and the authors get good mapping of the Hfq and sRNA interaction sites. Please comment on the necessity for trimming and how these RNAs are footprinted to the Hfq or sRNA interaction site in the protocol. The singleton footprints seem to be cleaner than RIL-seq data.

Minor comments.

The authors should mention and reference the Hi-GRIL-seq protocol in the introduction (lines 94-103). This technique is an inspiration for LiRIL-seq and its omission until late in the discussion seems out of place.

Line 357-358. The protocol is contrasted against RIL-seq and CLASH, both of these protocols are able to recover in vivo reactions and identify relevant base-pairing regions.

The authors recover statistically significant motifs with 100% of target mRNAs. Can the authors clarify if this is actually 100% of sRNAs with ≥ 6 interactions? The phrasing suggests that a motif is found for 100% of target mRNAs. What proportion of the dataset is represented by the 107 mRNAs (RNA2) used here?

Line 360-363. There is no evidence presented that LiRIL-seq reduces promiscuous ligation. An alternative view is that ligation within the complex, concentrated environment of the cell would be expected to result in more promiscuously ligation. Dilution of the sample in vitro will reduce promiscuous RNA-RNA interactions.

Line 363-364. Is there any evidence that there are less transient interactions recovered by LiRIL-seq (or more by RIL-seq?).

The protocol refers to 50 OD as the amount of biomass. While this is a standard unit, it is not widely used in the literature and is poorly understood by users. Culture volumes with OD values would make the protocol easier to reproduce.

Reviewer #3 (Remarks to the Author):

In this study, the authors describe an elegant modification of the available RNA-RNA interactome mapping approaches (e.g. RIL-seq, CLASH, Hi-GRIL-seq). The data provided suggest that the method yields comparable results to those of the RIL-seq and CLASH approaches, with the advantage that the ligation step between neighboring RNAs is done in vivo instead of in-vitro. Interestingly, the authors found a novel sRNA, FadZ, that regulates the expression of outer membrane porins and is important for fatty acid metabolism. The results also highlight the concept that one mRNA (ompD) may be regulated by a large number of sRNAs. The manuscript is well-written and easy to read. Most of the results presented are convincing and of high quality. However, in some cases the statements of the authors are not supported by the data presented and should be rephrased or supported by additional data. Below are some comments and suggestions that could improve the manuscript:

Major comments:

1. LiRIP-seq resembles other RNA-RNA interactome mapping approaches such as RIL-seq and CLASH. In the LiRIP-seq, the ligation step and the Hfq-3XFLAG pull-down were switched in comparison to RIL-seq, and a couple of steps were removed (e.g. UV-crosslinking, protein digestion). However, one of the

strengths of the RIL-seq is its computational pipeline and to the best of my understanding (lines 543-561) the authors nicely adopted that pipeline. One may argue that LiRIP-seq is a modification of the RIL-seq approach and perhaps it would be better for the scientific community to name the method in the manuscript in a similar name (vRIL-seq?) for consistency and ease of use as was done in other cases (e.g. iCLIP, PAR-CLIP, HITS-CLIP).

2. To validate the results of LiRIP the authors nicely carried out analyses that were done in previous RIL-seq studies (Figures 1C, 1E, 1F, 2B, 3B-F, etc.). It will strengthen their data if the authors will mention that their analyses recapitulate what was previously found for RNA-RNA interaction networks.

3. Fig 3A and lines 200-205: The comparison of LiRIP-seq data with previous Salmonella RIL-seq data is of value. However, by the way the analysis was done it is hard to draw any conclusions as the authors did. The RIL-seq data collected by Matera et al. 2022 was at OD=2.0 whereas the authors compare it to all of their 3 data points. A more accurate comparison would be to compare LiRIP-seq and RIL-seq data collected at the same time point (at OD =2.0 that the authors have).

4. Line 258: Figure 4C: There is a discrepancy between the data presented in the upper gel and the Western analysis at the bottom. ArcZ does not seem to reduce OmpD levels at top but it does at the bottom. Please elaborate on this difference.

5. Fig 6B: The presented data is of low quality and hard to interpret. Consider repeating on the experiment. Also loading control is missing.

6. Lines 314-317: Can the authors discuss the importance of reducing these porins under fatty acid metabolism?

7. Lines 328-331: The meaning of these results is unclear. In crp mutant the levels of FadZ are low and there is a growth defect. If you overexpress FadZ there is also a growth defect. In lines 312-313 the authors claim that FadZ is induced under these conditions so one would assume it is beneficial for the bacteria? What is the meaning of the growth defect upon its overexpression? Additional experiments can help clarify this. For example, expressing FadZ from an inducible promoter and testing a range of induction concentrations for their effect on growth.

8. Lines 342-344 & Fig 6I: The presented model states that all sRNAs inhibit ompD expression. The data does not support inhibition of ompD by STnc1010, CpxQ and MicA (Fig 4C). I suggest to distinguish between sRNAs that the authors document their ability to reduce ompD levels and the ones that do not.

Minor comments:

1. Line 75: A reference (<https://doi.org/10.7554/eLife.62438>) for internal sRNA originating from ORF of other genes.

2. Lines 94-113: Consider addressing Hi-GRIL-seq in this section instead of in the discussion (lines 371-374) as the authors were inspired by their studies for the development of LiRIP-seq.

3. Perhaps rephrase time point "2+3h" (written for the first time in line 175) to "stationary phase" for the ease of read?

4. Line 240: R1 should be RNA1?

5. Figure 5I-K: What do the error bars represent for the control sample? My understanding is that the fold change is calculated for all the other bars based on the control sample that serves as a reference.

6. Line 260: The data presented supports only 10 sRNAs that regulate OmpD (and for most of them, direct base-pairing was not demonstrated).

7. Lines 348: The LiRIP-seq has advantages as the authors described but it will be good to also discuss some limitations or weaknesses. For example, what is the meaning of the difference in the number of S-chimeras in the different growth phases (~400 at 0.5, ~850 at 2.0, ~1700 at 2.0+3n)? or another point is that it may seem that there are fewer chimeras captured for primary sRNA transcripts (Table S5). Does it mean that there is a bias towards processed sRNAs?

8. Lines 411-412: Add reference for UhpT3UTR that is called UhpU (<https://doi.org/10.7554/eLife.87151.1>).

9. Please keep sRNA names consistent across the manuscript. For example, CpxQ is used in the manuscript but in the tables it is referred to as STnc870. It can be confusing for researchers outside of the field.

Response to Reviewers' Comments:

Reviewer #1 (Remarks to the Author):

Small regulatory RNAs (sRNAs) play a crucial role in post-transcriptional gene regulation in bacteria, and control a wide range of cellular processes, including metabolic pathways, stress responses, quorum sensing, and virulence factor production. The bacterial RNA interactome ensures a rapid response to environmental change via both trans-acting and cis-acting mechanisms that usually involve base pairing with mRNAs to generate regulatory feedback loops and networks.

Until now, the study of the bacterial RNA-RNA interactome has involved laborious indirect methods that involve in vitro cross-linking and RNA ligation. Here, Liu et al report a novel approach that involves the pulse over-expression of RNA ligase to reveal the RNA-RNA interactome within living bacterial cells. Following extensive validation, the authors used the LiRIP-seq approach to investigate the dynamics of sRNA-mRNA interactions during bacterial growth. The authors report the discovery of a previously unknown regulatory role for the FabZ sRNA, which directly influences growth on fatty acids. The LiRIP-seq data also provide functional insights for a range of other functionally-uncharacterized sRNAs.

The paper is likely to be highly cited because it reports the RNA-RNA interactome of the model pathogen Salmonella enterica serovar Typhimurium and the role of 30 hub mRNAs that interact with > 4 sRNAs.

Our current understanding of the RNA-RNA interactome of Salmonella is based on 20 years of intensive research. Perhaps a more important impact of the Liu et al's study is that LiRIP-seq could be used to comprehensively catalogue sRNA-mRNA interaction networks in a wide range of bacterial species, without the need for decades of research.

The methods and experimental strategies used in the study were of a particularly high standard. The only significant change that I recommend is to the names given to the three stages of growth used to generate data for Figures 2 and 3. Rather than terming these as "OD 0.5", "OD 2.0" and "OD 2.0 + 3h", I suggest that names are used to build on approaches previously defined for Salmonella enterica serovar Typhimurium by the Vogel and Hinton laboratories.

The "OD 2.0" condition has been extensively used to define Salmonella Typhimurium gene regulation, and is usually referred to as "ESP" (= early stationary phase). The "OD 0.5" and "OD 2.0 + 3h" conditions do not yet have an accepted abbreviation. I suggest that "OD 0.5" is termed "EP" (Exponential Phase), and "OD 2.0 + 3h" is termed "SP" (Stationary Phase) in Figures 2 and 3, and throughout the paper. These abbreviations should be carefully defined in the Materials and Methods (line 492).

Response: We are very grateful to this reviewer for the positive feedback and euthanasic

support on our study! Following the major suggestion, we have changed the names of conditions to “EP”, “ESP” and “SP” in all the figures and throughout the paper.

Minor corrections

Line 110: Change “artifact” to “artefact”.

Response: Corrected.

Line 312: Add details of the media recipe containing octanoic acid. Add product codes for the oleic acid and octanoic acid fatty acids.

Response: We have added the detailed information in the revised manuscript (Materials and Methods).

Lines 370-371: Rephrase sentence to improve clarity.

Response: We have revised this sentence for better clarity.

Lines 389 - 404: Paragraph should be improved by avoiding the use of words such as “this”, “them”, “those” and “their” which introduce ambiguity to some sentences. e.g. at Line 394, what does “their” refer to?

Response: We have revised the paragraph to avoid ambiguity.

Lines 486 onwards (Materials and Methods): more details are required to allow experiments to be repeated. Manufacturers and product codes should be provided for all media components (Lennox broth, M9CA medium etc.

Response: Thanks for the reviewer’s kind suggestion. We have added the more detailed information in the revised manuscript (Materials and Methods).

Line 492: Change “Lenox” to “Lennox”.

Response: Corrected.

Line 513: explain the phrase “50 OD” in more detail.

Response: A culture volume corresponding to 50 OD (e.g., 100 ml for cells at OD₆₀₀ of 0.5) of bacterial cells were collected by centrifugation at 12,000 g for 5 min. We have included the detailed explanation in the revised manuscript (Materials and Methods).

Line 549: Explain the term “mates” in more detail.

Response: The “two mates” represents the 25 nt from both ends of a sequenced fragment. We have included this explanation in the revised manuscript.

Table S1: explain “IG::cat” in more detail.

Response: This strain was constructed in the Bossi lab from a previous study (Figueroa-Bossi et al. Genes Dev. 2009,23:2004-15, PMID: 19638370). “(rluC-rne) IG::cat” refers to that a cat cassette was inserted into the intergenic region between the rluC and rne genes as a selection marker and isogenic control for the rne-3071 (TS) strain.

Table S2: Explain the recombinant DNA manipulations in this table in more detail. For example, “via XbaI” lacks the detail needed to repeat the plasmid construction.

Response: Thanks for the suggestion. We have now provided detailed information for plasmid constructions in the Table S2.

Figure 1C: the terms EV and T4 need to be defined in the legend.

Response: Corrected. We have included in the legend. EV: empty vector. T4: pBAD-*t4rn11* (pYC582).

Figure 1D legend: add a brief explanation of what an “s-chimera” is.

Response: We have included a brief description in Figure 1B legend.

Figure 2C and Figure 3A: Some fonts used in the Venn diagram should be changed from black to white to improve legibility.

Response: Corrected.

Line 950: Add a brief explanation of what the “Hfq-LiRIP” data refers to.

Response: Corrected.

*Figure 5A: Check the annotation of the chromosomal region carefully. Both the STM3980 and STM3981 genes should be located between the *fre* gene and the *fadZ* gene?*

Response: We thank the reviewer’s careful reading. We have checked the annotations and confirmed that only gene STM3980 (1824 bp) is located between the *fre* gene and the *fadZ* gene in the SL1344 strain of *Salmonella enterica* serovar Typhimurium. However, in the LT2 strain pointed out by the reviewer, STM3980 is split into two genes (STM3980, 714 bp and STM3981, 1059 bp). We have updated the information in Figure 5A according to the annotations in SL1344 (the strain under study here).

Figure 6H and Figure S7F: Change “oleic acids” to “oleic acid”.

Response: Corrected.

Line 1006: Change “oleic acids” to “oleic acid”.

Response: Corrected.

Reviewer #2 (Remarks to the Author):

The authors present a modified version of the RNA-proximity dependent ligation techniques HiGRIL-seq and RIL-seq for profiling sRNA-mRNA interactions associated with Hfq. The major innovation is that library preparation is significantly simplified which is a major advantage as the current protocols – RIL-seq and CLASH have over 100 steps. This should facilitate broader use of the technique for sRNA interaction profiling. The disadvantage is that protocol does not include trimming or RppH treatment. A native 5'P end is required to facilitate ligation and the dataset is biased towards the subset of processed sRNAs that have a native 5'P and 3'OH end closely associated in vivo. Overall, the paper is thorough, very well written, and will be a valuable addition to the sRNA tool kit.

Response: We thank this reviewer to point out the value of our streamlined method, including its major strength as well as potential bias, which we have addressed below in greater details.

Major comments.

1. It would be useful to clarify the bias in the protocol. Lines 225-237 discuss the preference for processed and 3'UTR sRNAs. The title for this section seems misleading given the authors outline a bias against primary sRNAs. Please include a comparison of RIL-seq and LiRIL-seq ligation at native 5' ends. How biased is the protocol to native 5'P ends? Does the mRNA 3' end have a similar bias and are CDS/3'UTR interactions similarly under represented? What is the distribution of interaction distances from the 5' end and 3' end for LiRIL-seq and RIL-seq or CLASH?

Response: We thank this reviewer for the valuable comment. It is of high interest to us and the community to identify the systemic biases in the mainstream methodologies. To fully address the reviewer's question, we have performed a more detailed comparison between LiRIP-seq data (at OD2.0) and the published RIL-seq data (at OD2.0, PMID: 27588604).

First, analyzing the proportion of processed sRNAs among S-chimeras from LiRIP-seq or RIL-seq discovered that both RIL-seq and LiRIP-seq may have a preference to capture processed sRNAs in S-chimeras (see figure below or Fig. S3 A). The proportion of processed sRNAs in singleton fragments is considerably low (~7%), but is enriched for 7.6-fold in S-chimera in LiRIP-seq, and for 5.2-fold in S-chimera in RIL-seq at OD 2.0, respectively. This result suggests that processed sRNAs with 5'P are intrinsically prone to ligation by the T4 RNA ligase 1, regardless of the method used. Compared to RIL-seq, LiRIP-seq only has a slightly higher potential to ligate processed sRNAs in the absence of any trimming or end repairing steps *in vivo*.

Figure S3A. Comparison of the proportion of processed and primary sRNAs in S-chimera fragments between LiRIP-seq and RIL-seq.

Second, meta-analysis of S-chimera reads distribution in sRNAs showed that the 5'P ends of processed sRNAs are captured as S-chimeras in LiRIP-seq, whereas a non-uniform coverage was observed in S-chimeras in RIL-seq data (perhaps reflecting a trimming effect, see Figure below). For primary sRNAs, LiRIP-seq and RIL-seq performed equally well. Both approaches captured S-chimeras that are mapped at internal regions in sRNAs, consistent with our original observation in Figure 3H. Despite the observed enrichment of processed sRNAs in S-chimeras, a large proportion of chimeras were derived from primary sRNAs (70%, 43% and 29% at three growth conditions, respectively, Figure 3G). Thus, LiRIP-seq has reliably captured interactions for both primary sRNAs and processed sRNA, which is the main message we wish to highlight in the section title.

Distribution of sRNA fragments in S-chimeras over all processed and primary sRNAs. Each sRNA was divided in 100 bins and the number of sRNA fragments that mapped to each bin was calculated. A, Distribution of sRNA fragments in S-chimeras over processed sRNAs. B, Distribution of sRNA fragments in S-chimeras over primary sRNAs.

Finally, we have also calculated the distribution of sRNA targets in S-chimeras over the mRNA transcripts (See Figure below, or Fig. S3 E-F). Fragments of sRNA targets were enriched at 5'UTRs near the translational start codons, in line with canonical sRNA-mRNA regulatory mode in which sRNAs tend to bind around the ribosomal binding sites (RBS). This enrichment further indicates that most sRNAs were ligated to the 5' part of mRNAs, similarly shown by other approaches such as RIL-seq.

Figure S3 E-F, Distribution of sRNA targets fragments in S-chimeras over all protein-coding genes. (Left) Each gene was divided in 100 bins. The number of targets fragments that mapped to each bin was calculated. **(Right)** Distribution of sRNA targets fragments in S-chimeras at start codons. Dashed line indicates the position of start codons.

2. A more comprehensive comparison to the RIL-seq datasets would convince the reader that the streamlined protocol recovers equivalent information (or identify areas to be aware of differences). Comparisons of metrics like interaction strength, S-chimeras/interaction, and # S-chimeras recovered would help demonstrate comparable data. This data is available for *Salmonella* and *E. coli*. Figure 3A seems to focus on LiRIL-seq recovered interactions rather than comparing the techniques.

Response: Thanks for the reviewer's useful suggestions. We have reanalyzed the data at OD 2.0 from LiRIP-seq data and the RIL-seq data, to offer a more direct comparison between two techniques (Fig. 3A). For the Top10 sRNAs in benchmark, our analysis shows that LiRIP-seq captured a similar number of interactions (573) compared to RIL-seq (688), and that roughly a third of interactions were reproducibly captured by two different techniques. Therefore, we argue that LiRIP-seq performed equally well as RIL-seq, despite a highly streamlined workflow.

Figure 3A. Comparison of the sRNA-mRNA interactions found by LiRIP-seq and RIL-seq at the same condition (OD2.0). The bars indicate the numbers of predicted targets for 10 sRNAs that have most targets predicted. Venn diagram (inset) shows the overlap of all predicted targets for these 10 sRNAs between the two datasets.

In addition, we have compared several other metrics including interaction strength, number of S-chimera reads and number of S-chimeric interactions. As expected, RNA-RNA interactions obtained by LiRIP-seq have strong pairing strength, with significantly lower free energy compared to randomly shuffled RNA pairs (p -value= 4.36×10^{-14} , Kolmogorov-Smirnov test). Comparing the interaction strength between LiRIP-seq and RIL-seq, we found that LiRIP-seq showed a small but significant higher hybridization potential (p -value = 0.00385, Kolmogorov-Smirnov test). As to the other two metrics, very similar numbers of chimeric reads (per million reads) and chimeric interactions (per million reads) were detected by LiRIP-seq and RIL-seq (See Figure below, or Figure S3 B-D). Thus, our streamlined protocol recovers equivalent information as the published RIL-seq approach.

Figure S3 B-D. Comprehensive comparison between LiRIP-seq and RIL-seq.

Left Panel (Fig. S3D), Comparison of RNA-RNA interaction strength (ΔG , kcal/mol) for all RNA-RNA interactions in S-chimeras. Energies of RNA-RNA interactions were calculated by RNADuplex (PMID: 22115189). Pairs of interacting RNAs in LiRIP-seq were randomly shuffled. A two-sided Kolmogorov-Smirnov test was used to calculate p -values. **Middle**

Panel (Fig. S3B), Comparison of S-chimeric fragments detected in per million mapped reads. Right (Fig. S3C), Comparison of S-chimeras numbers detected in per million mapped reads.

3. HiGRIL-seq includes RppH treatment to remove the 5'PPP cap from RNAs to make them ligation compatible. Can the authors comment on whether this is a useful future addition to the protocol (ie: does HiGRIL-seq recover 5'PPP ends?).

Response: We thank this reviewer for the valuable comment. In the GRIL-seq paper (Han et al, Nature Micro, 2016. PMID: 28005055), the authors found that the ligation products of sRNA-targets were strongly reduced in *rppH* deletion strain, indicating RppH may generate more 5'P ends for ligation. We agree that an RppH treatment may be an interesting addition to the LiRIP-seq protocol in the future.

4. The RNAs recovered by LiRIL-seq are surprisingly short given at there is no trimming of RNAs (eg: Figure 1B), and the authors get good mapping of the Hfq and sRNA interaction sites. Please comment on the necessity for trimming and how these RNAs are footprinted to the Hfq or sRNA interaction site in the protocol. The singleton footprints seem to be cleaner than RIL-seq data.

Response: Both LiRIP-seq and RIL-seq protocols use the standard RNAtag-Seq method (Melamed et al, Nat Protoc, 2018; Shishkin et al, Nat Methods, 2015) to construct cDNA libraries. The library construction includes an RNA fragmentation step (mild alkaline hydrolysis), which fragment long transcripts into short fragments. Besides RNA fragmentation, several other factors might contribute to the observed shorter reads, such as RNA decay/trimming by endogenous ribonucleases during cell lysate preparation and preferential purification of the short fragments by the small RNA-seq protocol (2.5x RNAClean XP beads with 1.5x isopropanol). For a majority of sRNAs that bind to mRNA at 5' and trigger mRNA decay, we speculate that trimming may not be necessary; but it might help in theory to identify other noncanonical types of sRNA-target interactions such as 3' targeting, gene activation and sponges.

Minor comments.

The authors should mention and reference the Hi-GRIL-seq protocol in the introduction (lines 94-103). This technique is an inspiration for LiRIL-seq and it's omission until late in the discussion seems out of place.

Response: Thanks for the suggestion. We have moved the relevant part to the Introduction.

Line 357-358. The protocol is contrasted against RIL-seq and CLASH, both of these protocols are able to recover in vivo reactions and identify relevant base-pairing regions.

The authors recover statistically significant motifs with 100% of target mRNAs. Can the authors clarify if this is actually 100% of sRNAs with >=6 interactions? The phrasing suggests that a motif is found for 100% of target mRNAs. What proportion of the dataset is represented by the 107 mRNAs (RNA2) used here?

Response: These 107 different RNA2 contains 65 sRNAs, 18 3'UTRs, 15 CDSs, five IGRs

and four 5'UTRs. For the motif analysis, we've selected all the sRNAs (20 in total) with more than six (>6) different putative mRNA targets, most of which were located in RNA1 but also in RNA2. We've collected a total of 1577 target sequences, 1495 (94.80%) of which contain complementary sequence motifs. We have revised these numbers in the manuscript.

Line 360-363. There is no evidence presented that LiRIL-seq reduces promiscuous ligation. An alternative view is that ligation within the complex, concentrated environment of the cell would be expected to result in more promiscuously ligation. Dilution of the sample in vitro will reduce promiscuous RNA-RNA interactions.

Response: We agree that we do not have evidence to support our initial view or the alternative view. We have removed the discussion on reducing promiscuous ligations.

Line 363-364. Is there any evidence that there are less transient interactions recovered by LiRIL-seq (or more by RIL-seq?).

Response: Thanks. We have now clearly stated in the revised manuscript that this was our speculation.

The protocol refers to 50 OD as the amount of biomass. While this is a standard unit, it is not widely used in the literature and is poorly understood by users. Culture volumes with OD values would make the protocol easier to reproduce.

Response: Thanks for the suggestion. We have clarified this in the paper: a culture volume corresponding to 50 OD of bacterial cells were collected (e.g., 100 ml for cells at OD₆₀₀ of 0.5).

Reviewer #3 (Remarks to the Author):

In this study, the authors describe an elegant modification of the available RNA-RNA interactome mapping approaches (e.g. RIL-seq, CLASH, Hi-GRIL-seq). The data provided suggest that the method yields comparable results to those of the RIL-seq and CLASH approaches, with the advantage that the ligation step between neighboring RNAs is done in vivo instead of in-vitro. Interestingly, the authors found a novel sRNA, FadZ, that regulates the expression of outer membrane porins and is important for fatty acid metabolism. The results also highlight the concept that one mRNA (ompD) may be regulated by a large number of sRNAs. The manuscript is well-written and easy to read. Most of the results presented are convincing and of high quality. However, in some cases the statements of the authors are not supported by the data presented and should be rephrased or supported by additional data. Below are some comments and suggestions that could improve the manuscript:

Major comments:

1. LiRIP-seq resembles other RNA-RNA interactome mapping approaches such as RIL-seq and CLASH. In the LiRIP-seq, the ligation step and the Hfq-3XFLAG pull-down were switched in comparison to RIL-seq, and a couple of steps were removed (e.g. UV-crosslinking, protein digestion). However, one of the strengths of the RIL-seq is its computational pipeline and to the best of my understanding (lines 543-561) the authors nicely adopted that pipeline. One may argue that LiRIP-seq is a modification of the RIL-seq approach and perhaps it would be better for the scientific community to name the method in the manuscript in a similar name (vRIL-seq?) for consistency and ease of use as was done in other cases (e.g. iCLIPPAR-CLIP, HITS-CLIP).

Response: Excellent suggestion! Our method has combined experimental and computational features from GRIL-seq, RIP-seq and RIL-seq. It has been called with different interim names such as RIP-GRIL-seq and Fast-RIL before LiRIP-seq. Considering the benefit for a greater scientific community, we agree that it is a good idea to find a name that is easy to use and is consistent with the widely-appreciated RIL-seq terminologies in the prokaryotic RNA field. Thus, we propose to name our method iRIL-seq (intracellular RIL-seq), showing respect to the RIL-seq approach and also reflecting a major step of method evolution.

2. To validate the results of LiRIP the authors nicely carried out analyses that were done in previous RIL-seq studies (Figures 1C, 1E, 1F, 2B, 3B-F, etc.). It will strengthen their data if the authors will mention that their analyses recapitulate what was previously found for RNA-RNA interaction networks.

Response: Thanks for the valuable comment. We have now mentioned in the revised manuscript that our analyses have recapitulated the results and findings from RIL-seq studies.

3. Fig 3A and lines 200-205: The comparison of LiRIP-seq data with previous Salmonella RIL-

seq data is of value. However, by the way the analysis was done it is hard to draw any conclusions as the authors did. The RIL-seq data collected by Matera et al. 2022 was at OD=2.0 whereas the authors compare it to all of their 3 data points. A more accurate comparison would be to compare LiRIP-seq and RIL-seq data collected at the same time point (at OD =2.0 that the authors have).

Response: Following the suggestion from this reviewer and the reviewer #2, we have performed a side-by-side comparison between two datasets at the same condition (OD2.0, see the new Fig. 3A). The results showed that LiRIP-seq recovered similar numbers of interactions and shared a quarter of the interactions with RIL-seq.

4. Line 258: Figure 4C: There is a discrepancy between the data presented in the upper gel and the Western analysis at the bottom. ArcZ does not seem to reduce OmpD levels at top but it does at the bottom. Please elaborate on this difference.

Response: We apologize for the confusion. ArcZ actually consistently reduced the OmpD levels in both the upper gel and the Western blot at the bottom. In lane 6 of the upper SDS-PAGE gel, there were two distinct bands (red arrows indicated) corresponding to OmpD and maybe OmpF, respectively, whereas only the OmpD band was detected by the western blotting. We have enlarged the size of asterisks to clearly mark the bands for OmpD in upper and low gels (Figure 4C).

Figure 4C. Verification of OmpD regulation by different sRNAs. The *Salmonella* WT strain containing an empty vector or sRNA overexpression plasmids was grown overnight in LB. Total proteins were analyzed by 12% SDS-PAGE. The gel was stained with Coomassie brilliant blue, or subjected to Western blotting using a polyclonal anti-OMP antiserum. $\Delta ompD$ served as a OmpD-null control. The OmpD bands are indicated by asterisks.

5. Fig 6B: The presented data is of low quality and hard to intraperate. Consider repeating on the experiment. Also loading control is missing.

Response: Thanks for the suggestion. We have repeated the experiment and included the new results with improved quality. The levels of OmpD can be observed clearly in the revised Figure 6B. Similar amount of total protein from same number of cells were loaded to the PAGE gel. Signals from other abundant proteins on the gel support equal loading.

6. Lines 314-317: Can the authors discuss the importance of reducing these porins under

fatty acid metabolism?

Response: We think that during fatty acid metabolism, FadL is the preferred porin because FadL is more efficient in the uptake of long-chain fatty acids (Maloy SR, et al, 1981, JBC, PMID: 7012142). The activation of FadZ to repress multiple abundant porins might give more space to FadL, thereby remodeling the bacterial membrane to facilitate the uptake and utilization of long-chain fatty acids. We have included this idea in the revised manuscript in the Discussion section.

7. Lines 328-331: The meaning of these results is unclear. In crp mutant the levels of FadZ are low and there is a growth defect. If you overexpress FadZ there is also a growth defect. In lines 312-313 the authors claim that FadZ is induced under these conditions so one would assume it is beneficial for the bacteria? What is the meaning of the growth defect upon its overexpression? Additional experiments can help clarify this. For example, expressing FadZ from an inducible promoter and testing a range of induction concentrations for their effect on growth.

Response: Thank you for the insightful questions and suggestions. CRP, as a global regulator of carbon metabolism and catabolite repression, is crucial for bacterial growth on non-preferred sugars other than glucose. CRP controls the largest regulon in bacteria, including multiple other sRNAs including Spf and CyaR. Therefore, we think that the growth defect of Δcrp is a global effect on carbon metabolism when oleic acid was the sole carbon source, and cannot be simply explained by the reduced level of FadZ sRNA.

We also agree that over-expressing FadZ from the constitutive pLlacO-1 promoter does not directly reflect the endogenous role of FadZ. We took the suggestion using the pBAD plasmid and induced FadZ expression by different concentrations of L-arabinose. Unfortunately, we found that L-arabinose affected bacterial growth in oleic acid medium, likely because L-arabinose is a more preferred carbon source than oleic acid. Therefore, we are unable to study FadZ-mediated changes using this experimental setup. Nevertheless, we think that a growth phenotype caused by overexpressing FadZ indicates a physiological function in carbon metabolism under the control of Crp. This is an interesting line of research that we are pursuing in our follow-up study, including the identification of more relevant target genes and the elucidation of mechanisms for how FadZ controls long-chain fatty acid metabolism.

8. Lines 342-344 & Fig 6I: The presented model states that all sRNAs inhibit *ompD* expression. The data does not support inhibition of *ompD* by *STnc1010*, *CpxQ* and *MicA* (Fig 4C). I suggest to distinguish between sRNAs that the authors document their ability to reduce *ompD* levels and the ones that do not.

Response: Thanks for the valuable comment. We have revised Fig 6I and distinguished the three sRNAs in question (*STnc1010*, *CpxQ* and *MicA*) from other 12 sRNAs that showed regulation of *OmpD*.

Minor comments:

1. Line 75: A reference (<https://doi.org/10.7554/eLife.62438>) for internal sRNA originating from ORF of other genes.

Response: Thank you. This reference has been added.

2. Lines 94-113: Consider addressing Hi-GRIL-seq in this section instead of in the discussion (lines 371-374) as the authors were inspired by their studies for the development of LiRIP-seq.

Response: Thanks for the good suggestion. Following the suggestion by this reviewer and Reviewer #2, we have introduced the Hi-GRIL-seq approach in the introduction.

3. Perhaps rephrase time point “2+3h” (written for the first time in line 175) to “stationary phase” for the ease of read?

Response: Thank you. Following the suggestion by this reviewer and also reviewer #1, we have redefined the three growth stages in the revised manuscript. “OD 0.5” is termed “EP” (exponential phase), “OD 2.0” is termed “ESP” (early stationary phase), and “OD 2.0+3h” is termed “SP” (stationary phase).

4. Line 240: R1 should be RNA1?

Response: Thanks for the reviewer’s careful reading. We have corrected this typo.

5. Figure 5I-K: What do the error bars represent for the control sample? My understanding is that the fold change is calculated for all the other bars based on the control sample that serves as a reference.

Response: In the GFP reporter experiment, three biological replicates were performed. For the control GFP, we used one replicate as a reference (set to 1) and calculated the relative levels of the other two replicates, as well as the relative levels of all the other samples. Thus, the error bars for the control GFP represents variations among three biological replicates (standard deviation).

6. Line 260: The data presented supports only 10 sRNAs that regulate OmpD (and for most of them, direct base-pairing was not demonstrated).

Response: Thank you for this suggestion. Our data indicate that OmpD is regulated by a total of 12 sRNAs (>10 sRNAs), including eight sRNA regulators found in this study and four previously established sRNA regulators (RybB, InvR, SdsR and MicC). Therefore, we argue that OmpD represents one of the largest mRNA hubs in bacteria (regulated by >10 sRNAs). We have updated the Figure 6I to depict all the OmpD-regulating sRNAs.

7. Lines 348: The LiRIP-seq has advantages as the authors described but it will be good to also discuss some limitations or weaknesses. For example, what is the meaning of the difference in the number of S-chimeras in the different growth phases (~400 at 0.5, ~850 at 2.0, ~1700 at 2.0+3n)? or another point is that it may seem that there are fewer chimeras captured for primary sRNA transcripts (Table S5). Does it mean that there is a bias towards processed sRNAs?

Response: Thanks for the valuable comment. We acknowledge that *in vivo* ligation and our method may also have limitations/weaknesses. For example, the *in vivo* expression of T4 RNA ligase needs to be activated by an inducer (L-arabinose), which may affect the transcriptome during induction. Besides Hfq, our method may need further validation for other RBPs such as ProQ and CsrA, where UV crosslinking may be necessary to stabilize RNA chimeras.

Regarding the increasing number of interactions entering stationary phase, we do not yet fully understand the biological meanings, but we can speculate that multiple factors may be involved, such as increased expression of many stress-induced sRNAs, increased occupancy of Hfq, and increased mRNA decay, etc. Importantly, similar increase of interactions was also observed in RIL-seq studies in *E. coli* (PMID: 27588604), therefore it is not an artefact caused by our method of *in vivo* ligation.

Regarding the preference to processed sRNAs, we have elaborated on this addressing the first question from reviewer #2 (please see above). Briefly, this is a systemic bias (of the T4 RNA ligase) that we discovered for our *in vivo* approach but also for the RIL-seq method. Despite the bias, we have captured a large number of interactions for primary sRNAs (upto 70% of S-chimeras) and processed sRNAs with a highly streamlined workflow, with equal performance compared to RIL-seq. While we could use this preference to identify novel processed sRNAs and their targets, it would be possible to include some other *in vivo* steps (such as RppH treatment) to improve the method in the

future.

8. Lines 411-412: Add reference for UhpT3UTR that is called UhpU (<https://doi.org/10.7554/eLife.87151.1>).

Response: Thank you. We have added this reference in the revised manuscript.

9. Please keep sRNA names consistent across the manuscript. For example, CpxQ is used in the manuscript but in the tables it is referred to as STnc870. It can be confusing for researchers outside of the field.

Response: Thanks for the reviewer's suggestion. We have gone through all the tables and revised the sRNA names (e.g. in Tables S5).

REVIEWERS' COMMENTS

Reviewer #1 (Remarks to the Author):

The authors have addressed all my comments extremely well. The revised version looks very good, I look forward to seeing it in print!

One minor correction to be made at Line 581:
Please explain what a "mate" is.

Reviewer #2 (Remarks to the Author):

The authors have sufficiently addressed my comments in their revision.

Reviewer #3 (Remarks to the Author):

In this revised manuscript the authors successfully addressed most of my concerns.

Response to Reviewers' Comments:

Reviewer #1 (Remarks to the Author):

The authors have addressed all my comments extremely well. The revised version looks very good, I look forward to seeing it in print!

One minor correction to be made at Line 581:

Please explain what a “mate” is.

Response: Thank you for the kind reminder. We have added a brief explanation in the relevant part of the Methods section: The paired reads obtained from paired-end sequencing by Illumina NovaSeq are considered two mates.

Reviewer #2 (Remarks to the Author):

The authors have sufficiently addressed my comments in their revision.

Response: Thank you again for your suggestions that helped strengthen the manuscript.

Reviewer #3 (Remarks to the Author):

In this revised manuscript the authors successfully addressed most of my concerns.

Response: Thank you very much for your critical input and support.